# The effect of combining antibiotics on resistance: A systematic review and meta-analysis

Berit Siedentop[1,2]*, Viacheslav N Kachalov[2,3], Christopher Witzany[1], Matthias Egger[4,5,6], Roger D Kouyos[2,3†], Sebastian Bonhoeffer[1]*†

[1]Institute of Integrative Biology, Department of Environmental Systems Science, ETH Zürich, Zurich, Switzerland; [2]Department of Infectious Diseases and Hospital Epidemiology, University Hospital Zurich, University of Zurich, Zurich, Switzerland; [3]Institute of Medical Virology, University of Zurich, Zurich, Switzerland; [4]Institute of Social and Preventive Medicine (ISPM), University of Bern, Bern, Switzerland; [5]Population Health Sciences, University of Bristol, Bristol, United Kingdom; [6]Centre for Infectious Disease Epidemiology and Research, Faculty of Health Sciences, University of Cape Town, Cape Town, South Africa

*For correspondence:
berit.siedentop@env.ethz.ch (BS);
seb@env.ethz.ch (SB)

†These authors contributed equally to this work

Competing interest: The authors declare that no competing interests exist.

## eLife assessment

This is a methodologically state-of-the-art systematic review and meta-analysis of studies that addressed the question of whether the administration of multiple antibiotics simultaneously prevents antibiotic resistance development in individuals. The findings are **solid**. Rather than providing a precise answer, the synthesis of studies eligible for analysis leads to the conclusion that "our analysis could not identify any benefit or harm of using a higher or a lower number of antibiotics regarding within-patient resistance development." This article is **important** as it articulates the existing knowledge gap, but also serves as an example for careful future use of the meta-analysis methodology, when existing data just don't allow conclusions.

## Abstract

**Background:** Under which conditions antibiotic combination therapy decelerates rather than accelerates resistance evolution is not well understood. We examined the effect of combining antibiotics on within-patient resistance development across various bacterial pathogens and antibiotics.
**Methods:** We searched CENTRAL, EMBASE, and PubMed for (quasi-)randomised controlled trials (RCTs) published from database inception to 24 November 2022. Trials comparing antibiotic treatments with different numbers of antibiotics were included. Patients were considered to have acquired resistance if, at the follow-up culture, a resistant bacterium (as defined by the study authors) was detected that had not been present in the baseline culture. We combined results using a random effects model and performed meta-regression and stratified analyses. The trials' risk of bias was assessed with the Cochrane tool.
**Results:** 42 trials were eligible and 29, including 5054 patients, qualified for statistical analysis. In most trials, resistance development was not the primary outcome and studies lacked power. The combined odds ratio for the acquisition of resistance comparing the group with the higher number of antibiotics with the comparison group was 1.23 (95% CI 0.68–2.25), with substantial between-study heterogeneity ($I^2$=77%). We identified tentative evidence for potential beneficial or detrimental effects of antibiotic combination therapy for specific pathogens or medical conditions.

**Conclusions:** The evidence for combining a higher number of antibiotics compared to fewer from RCTs is scarce and overall compatible with both benefit or harm. Trials powered to detect differences in resistance development or well-designed observational studies are required to clarify the impact of combination therapy on resistance.

**Funding:** Support from the Swiss National Science Foundation (grant 310030B_176401 (SB, BS, CW), grant 32FP30-174281 (ME), grant 324730_207957 (RDK)) and from the National Institute of Allergy and Infectious Diseases (NIAID, cooperative agreement AI069924 (ME)) is gratefully acknowledged.

## Introduction

Antibiotics are one of the most significant advances in modern medicine, prescribed to treat various bacterial infections in both humans and animals and prevent infections, such as surgical site infections or opportunistic infections in immunocompromised individuals (*Taplitz et al., 2018*). However, this medical breakthrough is at risk due to the rising prevalence of antibiotic resistance and an inadequate pipeline of new antibiotics. This disturbing trend threatens to undermine the effectiveness of antibiotics and poses a severe challenge to public health worldwide (*Hutchings et al., 2019*; *Murray et al., 2022*). Hence, we need a more prudent use of antibiotics, and where antibiotics are needed, we need treatment strategies that reduce the risk that resistance emerges or spreads. Different strategies for the optimal use of antibiotics have been investigated theoretically and empirically (*Angst et al., 2021*; *Bliziotis et al., 2005*; *Paul et al., 2014*; *Tepekule et al., 2017*). Antibiotic combination therapy, i.e., the simultaneous administration of several antibiotics, is frequently discussed as a promising strategy for avoiding resistance evolution (*Angst et al., 2021*; *Bonhoeffer et al., 1997*; *Sullivan et al., 2020*; *Tepekule et al., 2017*; *Tyers and Wright, 2019*). Importantly, it is the standard of care for some bacterial pathogens, such as *H. pylori*, *Mycobacterium tuberculosis* (Mtb), or *Mycobacterium leprae* (*Alemu Belachew and Naafs, 2019*; *De Francesco et al., 2017*; *Singh et al., 2020*). However, it is unclear whether the effect of combination therapy on resistance is consistent for different pathogens.

There are several motivations for the use of antibiotic combination therapy, including to broaden the antibiotic spectrum in empirical treatment and reducing antibiotic resistance development (*Pletz et al., 2017*; *Roemhild et al., 2022*). The simultaneous occurrence of resistance mutations to multiple drugs is less likely than resistance to single drugs. Combination therapy should, therefore, reduce the development of resistance (*Bonhoeffer et al., 1997*). This expectation is supported by viral infections such as HIV, where multiple point mutations are required for resistance to combination antiviral therapy. However, it is less clear to what extent this reasoning extends to antibiotic therapy, where the same mechanism can facilitate bacterial survival against multiple antibiotics (*Du et al., 2018*; *Lázár et al., 2022*), and where horizontal transfer of resistance may occur. Indeed, the benefit of combining antibiotics for reducing resistance is debated for bacterial infections (*Holmes et al., 2016*). Using more antibiotics overall could lead to more resistance, as overall antibiotic consumption correlates with resistance (*Goossens et al., 2005*).

Two meta-analyses of randomised controlled trials (RCTs) comparing beta-lactam monotherapy to beta-lactam and aminoglycoside combination therapy found no differences in resistance development (*Bliziotis et al., 2005*; *Paul et al., 2014*). However, the effect of combining antibiotics on within-patient resistance development across many bacterial pathogens and various antibiotic combinations has not been addressed. Within-patient antibiotic resistance development, even if rare, may contribute to the emergence and spread of resistance. We performed a systematic review and meta-analysis to (i) test the effect of antibiotic combination therapy on within-patient resistance development and (ii) evaluate which factors affect the performance of combination therapy, as e.g., pathogen identity, treatment design and resistance assessment.

## Methods
### Inclusion criteria and search strategy

We did a systematic review and meta-analysis to summarise the evidence on the effect of antibiotic combination therapy on resistance development. We included RCTs and quasi-RCTs comparing treatments with a higher number of antibiotics to treatments with a lower number of antibiotics. Studies

were classified as quasi-RCTs if the allocation of participants to study arms was not truly random. We did not consider antiseptics or compounds supporting the activity of antibiotics, such as beta-lactam inhibitors as antibiotics itself. Whereas the antibiotic substances administered within one treatment arm had to be the same for all patients, the antibiotics could differ between treatment arms. We required baseline and follow-up cultures with resistance measurements to determine the treatment impact on resistance. We considered only antibiotic treatment regimens fixed for the period between two resistance measurements. Hence, we excluded sequential and cycling regimens.

We searched PubMed, EMBASE, and the Cochrane Central Register of Controlled Trials (CENTRAL) from inception up to 24.11.2022, using keywords, medical subject headings (MeSH), and EMTREE terms related to bacterial infection, antibiotics, combination therapy, resistance, and RCTs. We excluded complementary and alternative medicine and bismuth. The search strategy is detailed in Appendix 11. After a systematic deduplication process (*Bramer et al., 2016*), VNK (or CW) and BS independently screened the titles and abstracts, and, if potentially eligible, the full texts. Any discrepancies between VNK (or CW) and BS were discussed and resolved. At full-text screening, we excluded articles that were not accessible in English or German. We screened the references of eligible studies and the trials included in two previous meta-analyses (*Bliziotis et al., 2005*; *Paul et al., 2014*). We followed the PRISMA reporting guidelines (*Page et al., 2021*), the checklist is provided at: https://doi.org/10.17605/OSF.IO/GWEFY. We registered our protocol with PROSPERO (CRD42020187257).

## Outcomes

We used two definitions for the primary outcome resistance. A broader definition, 'acquisition of resistance,' and a stricter 'de novo emergence of resistance' definition, where the latter is a subset of the former. A patient was considered to have acquired resistance if, at the follow-up culture, a resistant bacterium (as defined by the study authors) was detected that was not present in the baseline culture. De novo emergence of resistance was defined as the detection of a resistant bacterium that was present at baseline but sensitive. Additional secondary outcomes included mortality from all causes and infection, treatment failure overall, treatment failure due to resistance, treatment change due to adverse effects, and acquisition/de novo emergence of resistance against non-administered antibiotics. The Appendix 9 provides further details.

## Data extraction and analysis

VNK (or CW) and BS independently extracted all study data using a standardised form (see https://doi.org/10.17605/OSF.IO/GWEFY). The data extracted included the proportion of patients who developed the two primary outcomes and the secondary outcomes and study characteristics such as type of trial (RCT or quasi-RCT), follow-up and treatment duration, number of antibiotics in the treatment arms, type of antibiotic, and presence of comorbidities. Any discrepancies in data extraction were discussed and resolved.

We calculated odds ratios (ORs) with 95% confidence intervals (CIs), comparing a higher with a lower number of antibiotics for each study. We combined ORs using a modified version of the Simmonds and Higgins random effects model (*Jackson et al., 2018*). If a study had more than two eligible treatment arms, they were merged for statistical analysis. Studies with zero events in both treatment arms were excluded from the statistical analysis. We used subgroup analyses and meta-regressions with multi-model inference to examine the influence of pre-specified variables on summary ORs. Variables included whether the antibiotic(s) used in the arm with the lower number of antibiotics are also part of the arm(s) with the higher number of antibiotics, the number of antibiotics administered, the age of the antibiotics (time since market entry), the administration of other non-antibiotic drugs, whether participants had specific comorbidities or were in intensive care, gram-status of the tested pathogens, and the length of antibiotic treatment and follow-up. We extended our predefined analysis regarding the reason for antibiotic treatment/type of pathogen, which was initially restricted to only *H. pylori* and Mtb, as we found enough studies to stratify by other conditions/pathogens. We, furthermore, performed post-hoc subgroup analyses to examine the following factors: treatment of resistant pathogens, additional antibiotic administration besides the fixed treatment, and the way of antibiotic administration (Appendix 6 section 2).

Between studies heterogeneity was estimated with $I^2$, using the criteria for $I^2$ specified in *Deeks et al., 2008* for classifying the degree of heterogeneity. CW and BS assessed each study's quality for

the main outcomes using the Risk of Bias tool (RoB 2, Appendix 3) (*Sterne et al., 2019*). To assess publication bias, we visually inspected the funnel plot and a modified Egger's test (Appendix 5). We performed sensitivity analyses on the model choice (Appendix 4 section 1), and risk of bias (Appendix 4 section 2), and performed a post-hoc trial sequential analysis (Appendix 8 section 2). Statistical analyses and visualisations were done in R (version 4.2.1) using packages *metafor* and *MuMIn* (*Bartoń, 2020*; *Viechtbauer, 2010*).

## Results

The search identified 3082 articles, which decreased to 1837 after deduplication. A total of 488 studies were eligible for full-text review, of which 41 studies qualified for inclusion. The screening of the citations of the 41 studies identified one additional eligible study (Appendix 11 section 2), for a total of 42 studies, 40 RCTs, and two quasi-RCTs, where the allocation method used is not truly random (*Figure 1*, *Supplementary file 1*). Twenty-nine studies could be included in the meta-analysis; 13 were excluded due to zero events in both treatment arms.

The included studies were published between 1977 and 2021, with a median publication year of 1995 and few recent studies (*Figure 2A*). The development of antibiotic resistance was typically not the main outcome: only nine studies (21%) explicitly defined a resistance outcome (*Supplementary file 1*, *Appendix 3—table 1*). Consequently, most studies did not have the statistical power to detect differences in within-patient resistance development even if we assume that the effect on resistance development is large between treatment arms (*Figure 2B*, Appendix 8). Twenty-two (52%) focused on a specific pathogen species (resistant *Acinetobacter baumannii*, *Escherichia coli*, *H. pylori*, Mtb, methicillin-resistant *Staphylococcus aureus* (MRSA), *Pseudomonas aeruginosa*, *Staphylococcus aureus*), or pathogen group (MAC, *Salmonella enterica* subsp. *enterica* serotype Thyphi, or *Salmonella enterica* subsp. *enterica* serotype Parthypi A).

The five most frequent reasons for antibiotic administration were treatment or prophylaxis of urinary tract infections (UTIs) (6 studies, 14%), MRSA (5 studies, 12%), *H*. pylori, MAC, and prophylaxis for hematological malignancy patients with four studies (10%), respectively. Twenty-three of the included studies (55%) compared treatment arms with at least one administered antibiotic in common; the remaining studies compared treatment arms with no overlap in administered antibiotics (*Supplementary file 1*). For the outcome acquisition of resistance, only two of all 42 studies had a low overall risk of bias according to the risk of bias assessment. Twelve (29%) were at high risk of bias, 28 (67%) at moderate risk of bias (Appendix 3).

The overall pooled OR for acquisition of resistance comparing a lower number of antibiotics versus a higher one was 1.23 (95% CI 0.68–2.25), with substantial heterogeneity between studies ($I^2$=77.4%). The latter OR was compatible with the OR for de novo emergence of resistance (pooled OR 0.74, 95% CI 0.34–1.59; $I^2$=77%). The overall pooled estimates are based on studies that focus on various clinical conditions/pathogens and compare different antibiotic treatments. To explore the impact of these and other potential sources of heterogeneity on the resistance estimates we performed subgroup analyses and meta-regression. The results for the two resistance outcomes are qualitatively comparable in the sense that individual estimates may differ, but show an overall similar absence of evidence to support either benefit, harm, or equivalence of treating with a higher number of antibiotics. Therefore, our focus in the following is on the acquisition of resistance (details on the emergence of resistance can be found in the Appendices 1–8).

Stratified analyses revealed that a higher number of antibiotics performed better than a lower number in the case of *H. pylori*, (pooled OR 0.14, 95% CI 0.03–0.55; $I^2$=41.7%, *Figure 3A*), and MAC (pooled OR 0.18, 95% CI 0.06–0.52; $I^2$=26.8%, *Figure 3A*), but worse in case of *P. aeruginosa* (pooled OR 3.42, 95% CI 1.03–11.43; $I^2$=1.54%, *Figure 3A*). Furthermore, a lower number of antibiotics performed better than a higher number if the compared treatment arms had no antibiotics in common (pooled OR 4.73, 95% CI 2.14–10.42; $I^2$=37%, *Appendix 6—table 1*), which could be due to different potencies or resistance prevalences of antibiotics (discussed in Appendix 6 section x.). In contrast, when restricting the analysis to studies with at least one common antibiotic in the treatment arms we found no evidence of a difference, only a weak indication that a higher number of antibiotics performs better (pooled OR 0.55, 95% CI 0.28–1.07; $I^2$=74%, *Figure 3B*). When considering only resistance measurements of antibiotics common to both treatment arms instead of all resistance measurements, the arm with a higher number of antibiotics shows a benefit in comparison to the one with

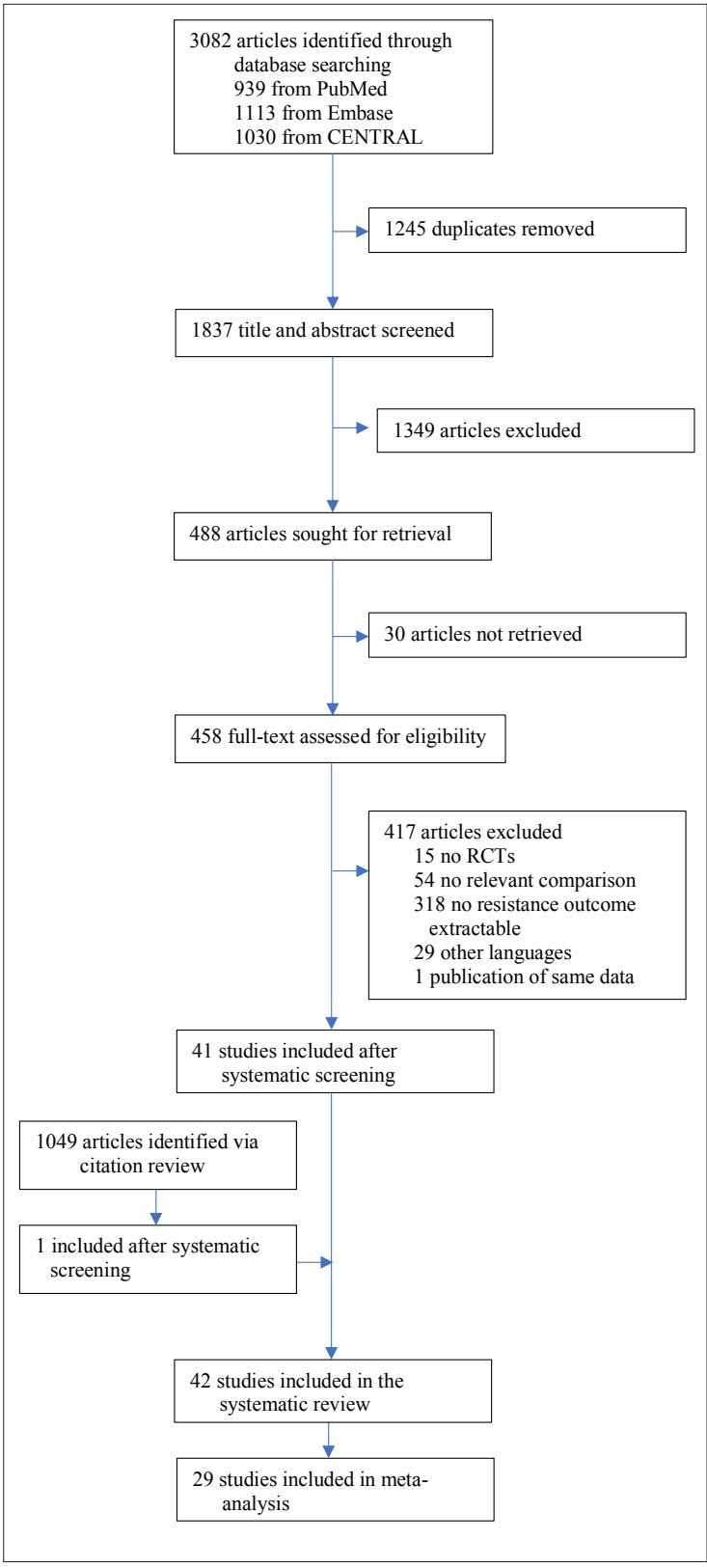

**Figure 1.** Study selection.

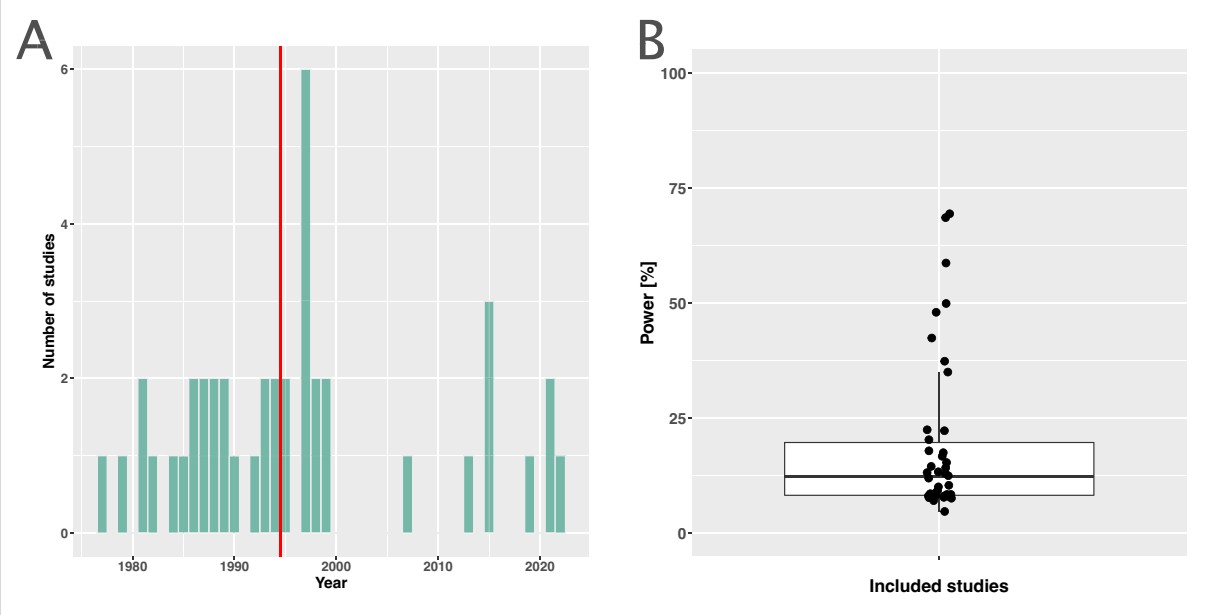

**Figure 2.** Measuring antibiotic resistance is not the current main objective of randomised controlled trials (RCTs). (**A**) Distribution of the publishing year of included studies. The number of studies published per year is shown, with the red vertical line indicating the median of the distribution. (**B**) Calculated statistical power [%] of included studies to detect an odds ratio of 0.5. The power calculations were based on equal treatment arm sizes. For the calculations, the treatment arm with the higher number of patients of the respective studies was used.

fewer (pooled OR 0.39, 95% CI 0.18–0.81; $I^2$=75%, Appendix 6 section viii.). If the study measured the acquisition of resistance of both gram-negative and positive bacteria, fewer antibiotics performed better (pooled OR 3.38, 95% CI 1.08–10.58; $I^2$=38.35%, Appendix 6 section vi.). Other sub-group analyses did not show any harm or benefit of using a higher number of antibiotics. The results for all subgroup analyses are presented in Appendix 6. The multi-model inference for our meta-regression showed that the only significant factor influencing the outcome acquisition of resistance is whether at least one common antibiotic was used in the comparator arms (for details see Appendix 7).

The inspection of the funnel plot and the modified Egger's test showed no indication of a publication bias (Appendix 5). The results were largely robust to the choice of the random effects model (Appendix 4). The probability of the secondary outcome 'alterations of the prescribed treatment due to adverse events,' was higher using more antibiotics in comparison to fewer (pooled OR 1.61, 95% CI 1.12–2.31; $I^2$=5%; Appendix 9 section v.). In 15 studies (36%), the proportion of patients with alterations of the prescribed treatment due to adverse events was reported, with three studies (20%) reporting zero cases in both treatment arms. All other analyses of secondary outcomes showed no indication of harm or benefit of treating with a higher number of antibiotics (Appendix 9).

## Discussion

We performed a meta-analysis of RCTs and quasi-RCTs not limited to a particular bacterial species, specific condition, or antibiotic combinations to assess the effect of antibiotic combination therapy on within-patient resistance development. Our analysis could not identify any benefit or harm of using a higher or a lower number of antibiotics regarding within-patient resistance development. However, we found some evidence that combining antibiotics may be beneficial or harmful for specific pathogens or infection types. Acquisition of resistance was rarely a primary objective of the included RCTs. Hence, they were typically not designed to detect differences in resistance development between treatment arms and underpowered for this endpoint. Therefore, the absence of evidence does not mean that there is convincing evidence for the lack of an effect of using more or fewer antibiotics on resistance development but rather highlights a knowledge gap. This is remarkable given that the general rise of resistance is an increasing concern (*Holmes et al., 2016*; *Murray et al., 2022*) and a priority area for health policy and public health (*Joshi, 2021*).

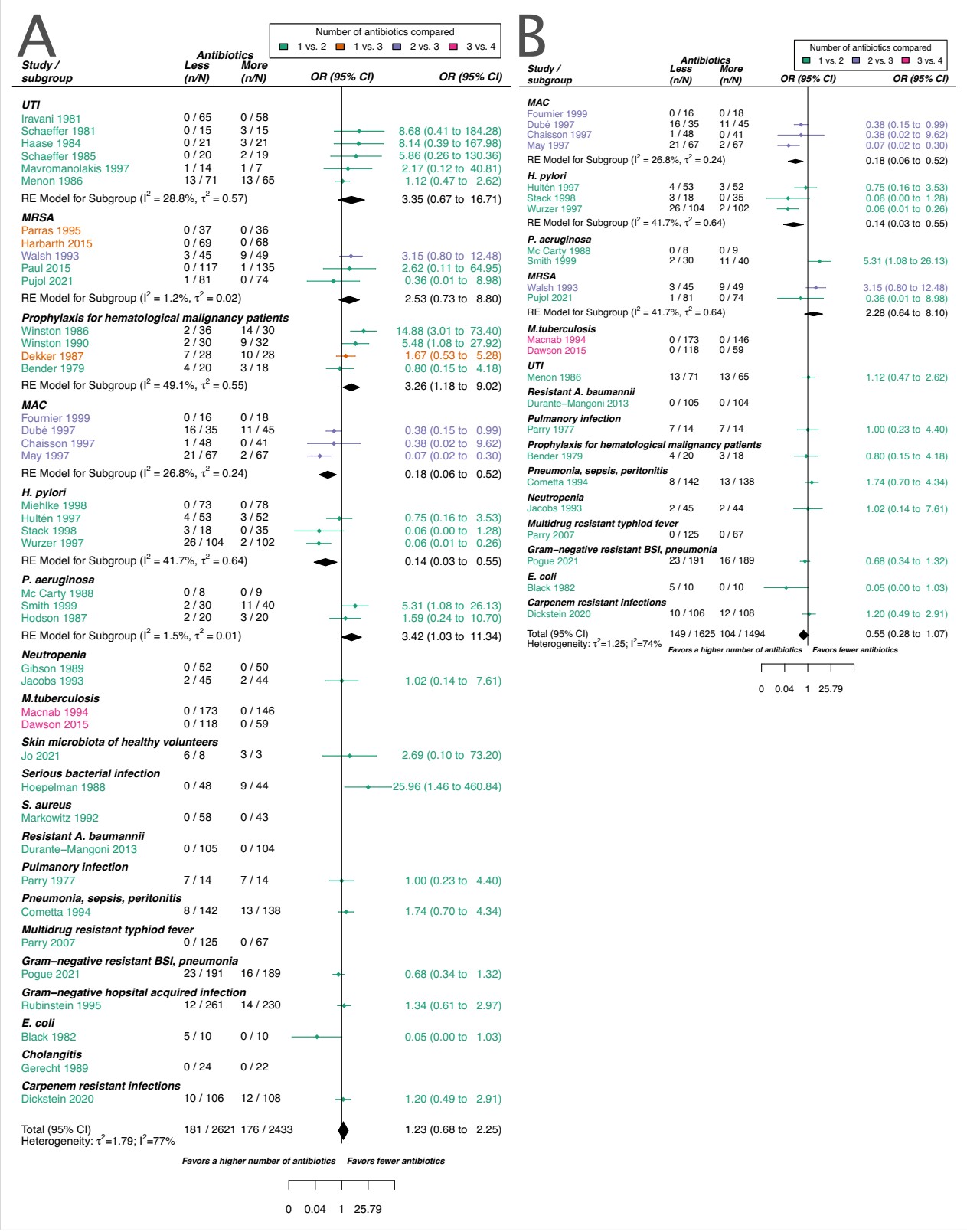

**Figure 3.** Forest plot of acquisition of bacterial resistance stratified by the reason antibiotics were administered. The colouring indicates the number of antibiotics that were compared in each study. (**A**) The overall pooled OR of all included studies. (**B**) The pooled OR of studies with at least one antibiotic in common in the treatment arms. UTI stands for urinary tract infection, methicillin-resistant Staphylococcus aureus (MRSA) for methicillin-resistant *Staphylococcus aureus*, MAC for *Mycobacterium avium* complex, and BSI for blood stream infection.

Our analysis showed that combining antibiotics reduced resistance development for *H. pylori* or MAC, in line with the current standard of care (*De Francesco et al., 2017*; *Kerantzas and Jacobs, 2017*). Surprisingly, we found only two studies that satisfied our inclusion criteria for Mtb (*Dawson et al., 2015*; *Macnab et al., 1994*), which may be considered the prime example of effective antibiotic combination therapy. The limited number of Mtb studies may be because antibiotic administration commonly varies during Mtb treatment, which conflicted with our inclusion criteria that necessitated a consistent treatment regimen for susceptibility measurements (Appendix 2). Both eligible Mtb studies were excluded from the analysis due to the absence of any events in either treatment arm.

Our main result, the absence of a general effect of combining antibiotics on resistance development, aligns with the two previous meta-analyses (*Bliziotis et al., 2005*; *Paul et al., 2014*). With 42 trials in our systematic review and 29 in the meta-analysis, our study provided a comprehensive assessment of the effect of antibiotic combination therapy on within-patient resistance. Whereas previous meta-analyses focused on a combination of specific antibiotic classes and included fewer than ten studies each, our study aimed to assess the general effect of combining antibiotics on resistance evolution across different bacterial pathogens. By including trials with different antibiotic combinations and bacterial pathogens, we increased clinical and statistical heterogeneity. We accounted for many sources of heterogeneity using stratification and meta-regression, but analyses were limited by missing information and sparse data.

Our findings have implications for the design of future studies of resistance development. Generally, the development of resistance within a patient is a rare event. However, even small differences could be relevant at the population level. To obtain reliable estimates of such differences and to better understand the factors influencing them, very large RCTs would be needed, which systematically investigate the development of antibiotic resistance and include resistance testing of each administered antibiotic. 19 (45%) of our included studies compared treatment arms with no antibiotics in common, and 22 studies (52%) had more than one antibiotic not identical in the treatment arms (*Supplementary file 1*). To better evaluate the effect of combination therapy, especially more RCTs would be needed where the basic antibiotic treatment is consistent across both treatment arms, i.e., the antibiotics used in both treatment arms should be identical, except for the additional antibiotic added in the comparator arm (*Supplementary file 1*). As such RCTs are costly and associated with high hurdles, the analysis of cohort studies could be an alternative approach. Over 25 years ago, *Fish et al., 1995* published a systematic summary of prospective observational studies reporting data on resistance development, including antibiotic combination therapy. Similarly, today, relevant cohort studies could be analysed collaboratively using various modern statistical methods to address confounding by indication and other biases (*Hernán, 2021*; *Hernán et al., 2022*). However, even with appropriate causal inference methods, residual confounding cannot be excluded when using observational data (*Schuster et al., 2023*). Therefore, RCTs will remain the gold standard to estimate causal relationships.

The main strength of this study is its comprehensive and systematic approach. For one, it allowed the identification of a knowledge gap regarding the effect of antibiotic combination therapy on resistance development. Furthermore, our study highlights several issues in the evidence base evaluating antibiotic combination therapy and resistance development. The included trials did not always test and report systematically the susceptibility against all administered antibiotics (*Supplementary file 1*). Some antibiotics might have had reduced potency or were ineffective due to pre-existing resistance mutations. Furthermore, in studies where treatment was not targeted against a specific pathogen, some antibiotics may have been inactive against the causative pathogen due to intrinsic resistance. Indeed, one of the reasons for using combination therapy is to broaden the bacterial spectrum for empirical therapy (*Roemhild et al., 2022*), which could contribute to an increased risk of antibiotic resistance spread.

Our study had several limitations. First, despite our systematic search, we might have missed relevant studies. Since resistance development is typically not a primary endpoint and often not reported systematically, relevant trials are challenging to identify. Our search strategy aimed to identify a broad range of trials considering resistance development. However, as a trade-off, our search strategy might have missed trials addressing a specific medical condition or drug combination. Second, our systematic review and meta-analysis included many older studies that did not follow the relevant reporting guidelines (*Schulz et al., 2010*), thereby hampering data extraction and potentially introducing bias.

Third, it is often challenging to discern the specific mechanisms by which resistance develops based on the data from clinical trials. This includes distinguishing whether resistance arises de novo, if the pathogen acquires resistance through horizontal gene transfer, if the patient becomes newly infected with a resistant pathogen, or if the pathogen was present but undetected at the beginning of treatment. These scenarios can impact the effectiveness of combination therapy. For example, combination therapy may be more likely to select any pre-existing resistant pathogens compared to monotherapy due to the use of multiple antibiotics. We addressed some of this heterogeneity by employing two different measures of resistance (Appendix 1). Furthermore, the variation in standards that classify bacteria as susceptible or resistant adds another layer of heterogeneity alongside the technical limitations in detecting resistance development.

In conclusion, combination therapy offers potential advantages and disadvantages regarding resistance evolution and spread. On the one hand, combination therapy typically increases the genetic barrier to resistance, and it has become the standard therapy for pathogens notorious for resistance evolution. Therefore, combination therapy remains a plausible candidate strategy to slow down resistance evolution. On the other hand, combination therapy generates selection pressure for resistance to multiple antibiotics simultaneously and could, therefore, accelerate resistance evolution – especially in the microbiome. Given the critical nature of this context, it is profoundly disconcerting that there is a lack of evidence elucidating the impact of combining antibiotics on the development of resistance.

## Acknowledgements

We thank Anthony Hauser, João Pires, and Frédérique Lachmann for helpful discussions and Annelies Zinkernagel and Johannes Nemeth for critical reading of the manuscript. Furthermore, we are grateful for all contacted study authors who responded to our inquiries and provided further information.

## Additional information

### Funding

| Funder | Grant reference number | Author |
|---|---|---|
| Swiss National Science Foundation | 310030B_176401 | Sebastian Bonhoeffer |
| Swiss National Science Foundation | 32FP30-174281 | Matthias Egger |
| Swiss National Science Foundation | 324730_207957 | Roger D Kouyos |
| National Institute of Allergy and Infectious Diseases | AI069924 | Matthias Egger |

The funders had no role in study design, data collection and interpretation, or the decision to submit the work for publication.

### Author contributions

Berit Siedentop, Conceptualization, Data curation, Software, Formal analysis, Validation, Investigation, Visualization, Methodology, Writing – original draft, Writing – review and editing; Viacheslav N Kachalov, Christopher Witzany, Data curation, Validation, Investigation, Writing – review and editing; Matthias Egger, Supervision, Writing – review and editing; Roger D Kouyos, Sebastian Bonhoeffer, Conceptualization, Resources, Supervision, Funding acquisition, Methodology, Writing – review and editing

### Author ORCIDs

Berit Siedentop  https://orcid.org/0000-0003-0802-7681
Viacheslav N Kachalov  http://orcid.org/0000-0002-0310-3764
Christopher Witzany  http://orcid.org/0000-0002-7128-6419
Matthias Egger  http://orcid.org/0000-0001-7462-5132
Roger D Kouyos  https://orcid.org/0000-0002-9220-8348

Sebastian Bonhoeffer ![ORCID] http://orcid.org/0000-0001-8052-3925

Reviewer #2 (Public Review): https://doi.org/10.7554/eLife.93740.3.sa1
Author response https://doi.org/10.7554/eLife.93740.3.sa2

## Additional files

### Supplementary files

• Supplementary file 1. Overview of the 42 randomised controlled trials (RCTs) or quasi-RCTs included in the systematic review and meta-analysis. The underlined antibiotics indicate that resistance measurements were made for this antibiotic, reported and extractable from the studies. Justification for resistance outcome extraction is given in *Appendix 3—table 1*.

• MDAR checklist

### Data availability

All data are contained in the manuscript, supplementary information or online at https://doi.org/10.17605/OSF.IO/GWEFY.

The following dataset was generated:

| Author(s) | Year | Dataset title | Dataset URL | Database and Identifier |
|---|---|---|---|---|
| Siedentop B, Kachalov VN, Witzany C, Egger M, Kouyos RD, Bonhoeffer S | 2023 | The effect of combining antibiotics on resistance: A systematic review and meta-analysis | https://doi.org/10.17605/OSF.IO/GWEFY | Open Science Framework, 10.17605/OSF.IO/GWEFY |

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

## Appendix 1

### Definitions of resistance development

To measure resistance development in patients with standard clinical routines is challenging. Without antibiotic pressure, a resistant strain might be present within the patient at low frequency and might not be detected with a culture due to detection limits. With antibiotic treatment, the frequency of this resistant strain might rise and, therefore, the strain might be detected in a follow-up culture. In this case, resistance did not develop de novo, but it is difficult to distinguish this case from an event where it did. Furthermore, the genetic relatedness is not always checked between initial and follow-up cultures, meaning that the resistant bacterium at a follow-up culture could have been also transmitted from a different body site or from other infection sources. To give a more comprehensive overview of how antibiotic treatment strategies might affect the resistance development, we, therefore, choose to present the results of two resistance estimates. A broader estimate, acquisition of resistance, and a stricter estimate de novo emergence of resistance, where the latter is a subset of the former. A patient is considered to have acquired resistance, if at the follow-up culture there has been a resistant (as defined by the study authors) bacterial species detected, that has not been detected in the baseline culture. A patient is considered to have de novo emergence of resistance, if at follow-up up culture a resistant bacterium was detected, that has already been detected at the baseline culture, but is sensitive. De novo emergence of resistance is nested in the definition of acquisition of resistance. In acquisition of resistance, we account for bacteria at low abundance that could have been already present at the beginning of treatment, but not detected at screening. In this definition, it is impossible to distinguish though, whether the bacteria already colonised the patient or whether the patient was newly infected by an external source during treatment and when the bacterium developed resistance. We also included the stricter definition of de novo emergence of resistance. For de novo emergence we only consider cases where a sensitive bacterium was cultured at baseline. In this definition it is less likely to count cases, where resistant bacteria were transmitted from an external source, as a de novo emergence event. But there are cases, which are counted as an event of de novo emergence of resistance, where in fact resistance did not develop newly, but resistance was only selected during treatment. This could be the case when a sensitive bacterium was cultured at baseline and the same kind of bacterium was also present at a non-detectable frequency as a resistant phenotype. Overall both resistance development definitions have their limitations and capture slightly different impacts of antibiotic treatment on resistance.

In the main manuscript, we showed results for the outcome acquisition of resistance and in the following section, the main pooled estimates for de novo emergence of resistance are presented.

## Appendix 2

### Main estimates for de novo emergence of resistance

As for the acquisition of resistance (main text *Figure 3*), we did not identify a difference in using a higher number of antibiotics in comparison to less if the de novo emergence of resistance is considered.

Counterintuitively, for *Mycobacterium tuberculosis* (Mtb) – which may be regarded as the flagship of antibiotic combination therapy – we could only identify two studies matching our inclusion criteria via our systematic search (main text *Figure 3*, *Appendix 2—figure 1*). Since the 1950s the administration of antibiotics often changes within the Mtb treatment period (*Fox et al., 1999*; *Kerantzas and Jacobs, 2017*). With the early establishment of changing antibiotics within the Mtb treatment period, it would be understandable, that resistance development measurements of periods with fixed antibiotic treatment, which is an inclusion criterion for our review, got less frequent over the years. Therefore, the relatively small proportion of Mtb studies included in our review is not surprising.

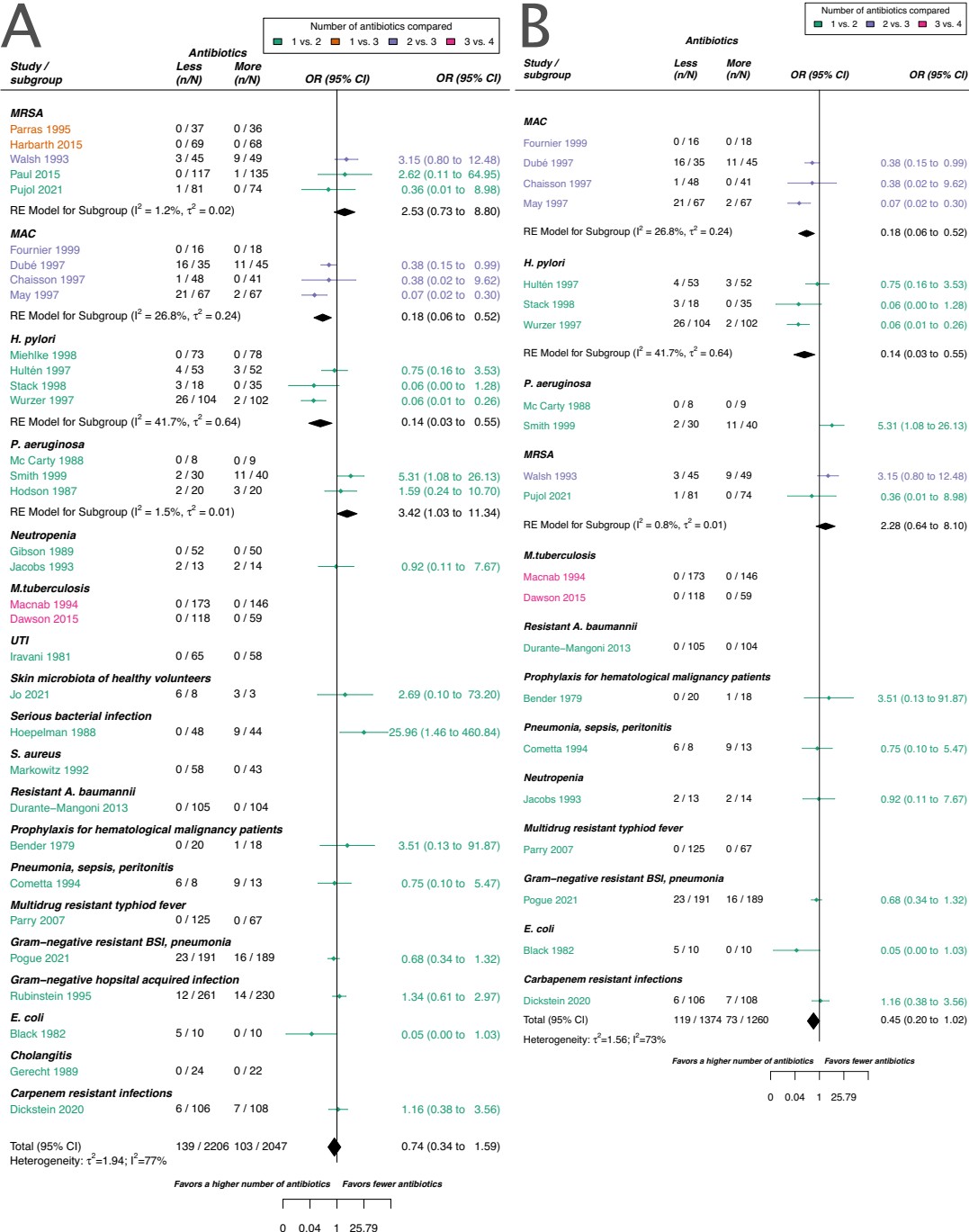

**Appendix 2—figure 1.** Forest plot of de novo emergence of bacterial resistance stratified by the reason antibiotics were administered. The colouring indicates the number of antibiotics that were compared in each study. (**A**) The overall pooled OR of all included studies. (**B**) The pooled OR of studies with at least one antibiotic in common in the treatment arms. MRSA stands for methicillin-resistant *Staphylococcus aureus*, MAC for *Mycobacterium avium* complex, and BSI for blood stream infection.

## 1. All studies

For all studies meeting our inclusion criteria and reporting data of de novo emergence of resistance, our estimate did not suggest a difference between using a higher number of antibiotics in comparison to less. This result was in line with our main outcome acquisition of resistance. Nevertheless, for de novo emergence of resistance, there was a slight trend observable which suggested a benefit of

using a higher number of antibiotics. However, we could not identify a clear benefit (pooled OR 0.74, 95% CI 0.34–1.59, *Appendix 2—figure 1A*). This trend might be due to the stricter definition of de novo emergence relative to the acquisition of resistance. In the definition of acquisition of resistance bacterial species that are different from the initial identified infecting organism are included, whereas for de novo emergence of resistance, they are not necessarily included. For de novo emergence of resistance, the efficacy of antibiotic treatment against the considered bacteria is, therefore, expected to be higher than for acquisition of resistance, as antibiotics typically have a specific bacterial spectrum of activity. The model including all studies reporting de novo emergence of resistance showed a substantial amount of heterogeneity ($I^2$=77%, *Appendix 2—figure 1A*).

## 2. Studies with at least one antibiotic common to both treatment arms

To compare more similar antibiotic treatments, we also estimated the effect of de novo emergence of resistance based on studies, that had at least one antibiotic common to the comparator arms. With this restriction, we also did not identify a difference between using a higher number of antibiotics in comparison to less, but we observed a stronger tendency of a benefit of using a higher number of antibiotics (pooled OR 0.45, 95% CI 0.20–1.02, *Appendix 2—figure 1B*). The model for studies reporting de novo emergence of resistance, and with at least one common antibiotic in the comparator arms showed still a substantial amount of heterogeneity ($I^2$=73%, *Appendix 2—figure 1B*).

## Appendix 3

## Risk of bias assessment

To assess the risk of bias for our two main outcomes we used the RoB 2 tool (*Sterne et al., 2019*). The results of the risk of bias assessments for acquisition, and de novo emergence of resistance differed only marginally, which can be explained by the overlap of those two definitions. We defined de novo emergence of resistance as a stricter subset of acquisition of resistance (Appendix 1). In both cases, two studies were classified overall with a low risk of bias, and about 50 % percent of the studies were classified overall with some concerns of bias (67% acquisition of resistance, 72% emergence of resistance, *Appendix 3—figure 1*). The highest source of at least some concern was the selection of the reported results. As the development of resistance is not a typical main objective of RCTs, and since we included a large proportion of rather old studies, the resistance outcome is often not well (pre-)defined (*Appendix 3—table 1*) and not presented in a systematic way, which can explain the risk of bias observed in the category 'selection of the reported results.' Since the studies were rather underpowered (main text: *Figure 2B*) to detect the resistance development, missing data was commonly a high risk of concern in the domain of 'deviations from intended interventions.' The detailed output of the risk of bias assessment using the RoB 2 tool can be found at OSF 'https://doi.org/10.17605/OSF.IO/GWEFY.'

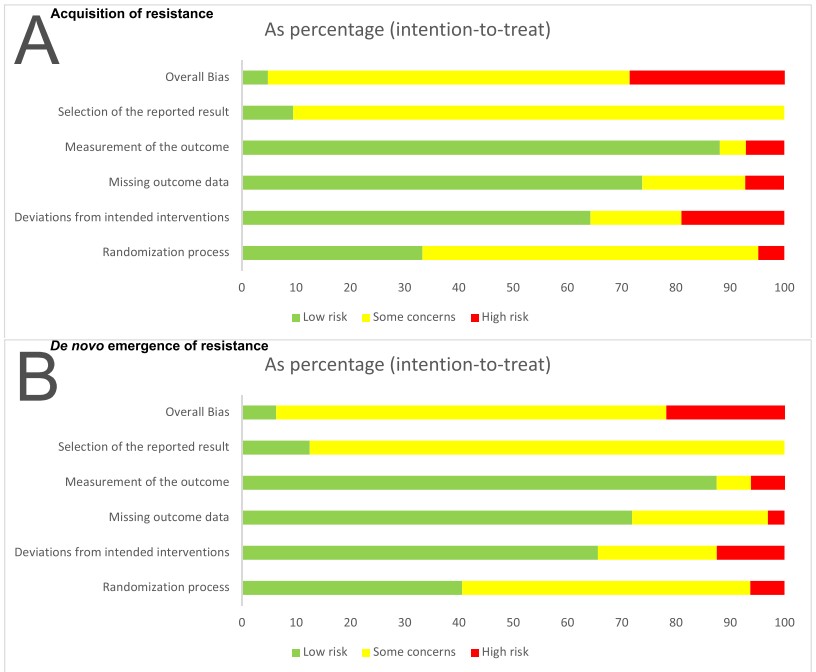

**Appendix 3—figure 1.** Risk of bias summary for the two main outcomes: (**A**) Acquisition of resistance, (**B**) de novo emergence of resistance.

**Appendix 3—table 1.** Justification for extraction of resistance development.
The definitions of resistance development are stated as given by the study authors. In case no explicit definition was given, we state a justification for extraction and indicate it with (*). Note that for data extraction for the publications of *Dickstein et al., 2020* and *Pogue et al., 2021* additional publications of the same studies were consulted by *Paul et al., 2018* and *Kaye et al., 2023*, respectively. Resistance breakpoints are stated in case numerical values were given in the respective studies. See *Supplementary file 1* for which antibiotics the studies tested and reported extractable resistance data.

| Study | Definition of resistance development given by study authors or justification for extraction |
|---|---|
| *Bender et al., 1979* | Susceptibility testing for gentamicin of the flora was performed at randomisation and twice weekly after with the Kirby-Bauer disk technique and microtiter minimal inhibitory concentration. (*) |

*Appendix 3—table 1 Continued on next page*

*Appendix 3—table 1 Continued*

| Study | Definition of resistance development given by study authors or justification for extraction |
|---|---|
| **Black et al., 1982** | Patients were infected with a known strain and all stool cultures and rectal swabs were plated and tested for trimethoprim resistance. (*) |
| **Chaisson et al., 1997** | Testing of isolates to susceptibility for clarithromycin, ethambutol, and clofazimine was performed before the entry of the study and monthly for 6 mo in broth by the method of Heifets.(*) |
| **Cometta et al., 1994** | All microorganisms were sensitive to imipenem at randomisation and follow-up cultures were performed. (*) |
| **Dawson et al., 2015** | Susceptibility testing at randomisation and for the following cultures by rapid testing. Susceptibilities to isoniazid, rifampicin, and fluoroquinolones were determined by line probe assay. (*) |
| **Dekker et al., 1987** | At admission, cultures were performed and surveillance cultures were done twice a week. Gram-negative bacilli were tested for antibiotic susceptibility. The minimal inhibitory concentrations were assessed by agar dilution technique. An MIC of ≥2 µg/mL was considered resistant for ciprofloxacin, an MIC of ≥4 µg/mL for trimethoprim, and an MIC ≥75 µg/mL for sulfamethoxazole. (*) |
| **Dickstein et al., 2020** | Development of a new colistin-resistant (ColR) isolates within 28 d from study enrolment. To be considered a new ColR isolate, the ColR isolate had to be detected on day 7 or later in patients for whom the baseline isolate was colistin-susceptible, and for whom no ColR isolate was cultured from the rectal swab taken on day 1. Susceptibility was determined by broth microdilution. Colistin resistance was defined as an MIC >2 mg/L. |
| **Dubé et al., 1997** | All available isolates were tested for susceptibility to clarithromycin. Patients were evaluated at the time of enrolment, 2 and 4 wk later, and then every 4 wk. Clarithromycin resistance was defined as detectable growth in a concentration of clarithromycin of 8 µg/mL. (*) |
| **Durante-Mangoni et al., 2013** | The identification of a colistin-resistant *Acinetobacter baumannii* during treatment was defined as resistance emergence. Resistance was determined by the microdilution method and/or E-test. |
| **Fournier et al., 1999** | Susceptibility testing was performed at study entry after 2 mo and classification was performed according to Heifets. (*) |
| **Gerecht et al., 1989** | Emergence of resistance was defined as one cause of treatment failure. Emergence of resistance was classified as the detection of an infecting microorganism resistant to more than 4 µg/mL of gentamicin sulfate or more than 128 µg/mL of mezlocillin sodium during treatment while the patient shows indications of cholangitis. |
| **Gibson et al., 1989** | Microbiological assessment of the blood was performed before treatment and 96 hr after treatment. (*) |
| **Haase et al., 1984** | Susceptibility was assessed before therapy, during therapy, and after therapy. Susceptibility testing was performed with disk dilution method, and agar dilution method. Resistance results were reported for reinfections defined as the reappearance of infection with a different organism after completion of therapy. Resistance against norfloxacin and trimethoprim-sulfamethoxazole was defined as a larger inhibition zone diameter of 0.17 and 0.16 mm, respectively, or/and a MIC larger than 16 µg/mL and 3.4–64 µg/mL, respectively. (*) |
| **Harbarth et al., 2015** | Susceptibility assessment was performed at baseline and at the end of treatment. Susceptibility was performed with a disc diffusion method phenotypically and genotypically. (*) |
| **Hodson et al., 1987** | *P. aeruginosa* had to be sensitive at inclusion and resistance was measured and reported after 10 d of treatment. Sensitivity was determined by standard disc methods. (*) |
| **Hoepelman et al., 1988** | Susceptibility was assessed before, during, and after treatment. Susceptibility testing was performed with the disc diffusion method and minimum inhibitor concentrations were assessed for blood cultures and patients with no response to treatment with the agar dilution technique. Resistance for the agar dilution technique was defined as an MIC of ≥32 µg/mL for ceftriaxone, ≥8 µg/mL for gentamicin, and ≥32 µg/mL for cefuroxime. For the disc diffusion method 30 µg ceftriaxone, 40 µg gentamicin, and 60 µg cefuroxime were used. If the zone of inhibition was ≤18 mm cultures were classified as ceftriaxone resistant and sensitive if the zone was ≥26 mm and intermediate in between. For gentamicin the values were ≤20 mm and ≥28 mm and for cefuroxime ≤20 mm and ≥28 mm, respectively.(*) |
| **Hultén et al., 1997** | Susceptibility was assessed by E-test at inclusion and 12 wk after treatment determination. (*) |
| **Iravani et al., 1981** | Susceptibility testing at baseline, during treatment and at follow-up. Testing was performed with Bauer's disc diffusion method using 30 µg nalidixic acid, 1.25 µg trimethoprim, and 23.75 µg sulfamethoxazole. (*) |
| **Jacobs et al., 1993** | Emergence of resistance was defined as treatment failure with resistance, i.e., bacteriological failure with the reisolation of original pathogen(s) resistant to the study antibiotic(s) after treatment. |
| **Jo et al., 2021** | Susceptibility testing before treatment and after treatment by culture. (*) |
| **Macnab et al., 1994** | Susceptibility testing before treatment and after around 90 doses. (*) |
| **Markowitz et al., 1992** | Susceptibility was assessed by microdilution method before treatment and for the last continuous positive culture during treatment. Furthermore, susceptibility was assessed for relapse isolates and isolates phenotypically different from the initial one. (*) |
| **Mavromanolakis et al., 1997** | Susceptibility was assessed before treatment, after 2 wk, at the end of treatment, and 2 wk after treatment by disk diffusion method. (*) |
| **May et al., 1997** | Susceptibility was assessed at treatment start, after 2 mo, and in case of relapse by the Becton Dickinson method. (*) |
| **McCarty et al., 1988** | Susceptibility was assessed at admission, every 4 d during treatment, and within 48 hr after treatment by broth microdilution method using the American Microscan Gram Negative-Panel. (*) |
| **Menon et al., 1986** | Susceptibility was assessed before therapy, and after 1 and 2 wk after therapy. (*) |

*Appendix 3—table 1 Continued on next page*

*Appendix 3—table 1 Continued*

| Study | Definition of resistance development given by study authors or justification for extraction |
|---|---|
| *Miehlke et al., 1998* | Susceptibility was assessed before and after treatment by E-test. An MIC of ≤0.125 mg/L was considered clarithromycin sensitive and an MIC of ≥2 mg/L resistant. An MIC of ≤2 mg/L was considered amoxicillin susceptible and an MIC of ≥4 mg/L resistant. (*) |
| *Parras et al., 1995* | Susceptibility was assessed at baseline and at the end of therapy by agar dilution method or automated microdilution methods. (*) |
| *Parry et al., 1977* | Susceptibility was assessed before, during, after treatment, after 2 wk, and after 6 mo after treatment by Bauer's method. (*) |
| *Parry et al., 2007* | Susceptibility was assessed before therapy and after treatment by E-test, disk diffusion method. Ofloxacin was tested by disk diffusion method with a 5 µg and organisms were declared susceptible with a breakpoint ≤2 µg/mL and resistant with a breakpoint ≥8 µg/mL. Azithromycin was also tested with the disk diffusion method (15 µg disk), but no clear breakpoints were defined. Instead, azithromycin was determined by E-test according to the manufacture's guidelines. (*) |
| *Paul et al., 2015* | Development of resistance was defined as the acquisition of *S. aureus* resistant to any of the study drugs or vancomycin-resistant *Enterococci*. |
| *Pogue et al., 2021* | Number of patients, who developed colistin resistance during therapy. Resistance was assessed with broth microdilution and declared as colistin-resistant with an MIC ≥4 mg/L. |
| *Pujol et al., 2021* | Emergence of resistance to studying drugs during treatment according to EUCAST. |
| *Rubinstein et al., 1995* | Resistance emergence was assessed by measuring MICs before, during, and after treatment. Disk diffusion testing was performed with disks of 30 µg ceftazidime, 30 µg ceftriaxone, and 10 µg tobramycin. An MIC ≤8 mg/L was considered susceptible for ceftazidime and ceftriaxone and a MIC ≥32 mg/L was considered resistant for ceftazidime and an MIC ≥64 mg/L for ceftriaxone. An MIC ≤4 mg/L was classified as susceptible for tobramycin, and an MIC ≥8 mg/L as resistant. |
| *Schaeffer et al., 1981* | Susceptibility was assessed before therapy, after 7 d, and after 5 to 9 d after therapy by plating. Susceptibility testing was performed by plating 0.1 mL of culture on Mac Conkey agar containing 100 µg/mL cinoxacin or 1–24 µg/mL trimethoprim-sulfamethoxazole. Any growing culture was considered resistant and resistance tests were confirmed with standard agar sensitivity testing to a maximum concentration of 100 µg cinoxacin or 80–400 µg trimethoprim-sulfamethoxazole. (*) |
| *Schaeffer and Sisney, 1985* | Susceptibility testing was performed before therapy, during therapy, and after 5 to 7 d after therapy by plating. 0.1 mL of cultures were plated on either Mueller-Hinton agar containing 10 µg/mL agar of norfloxacin or 1–24 µg/mL agar trimethoprim-sulfamethoxazole with 5% lysed red blood cells from the horse. Any growing culture was considered resistant and resistance tests were confirmed with tube dilution sensitivity testing to a maximum concentration of 100 µg/mL norfloxacin or 32–608 µg/mL trimethoprim-sulfamethoxazole. (*) |
| *Smith et al., 1999* | Susceptibility was assessed at inclusion, and at the end of treatment by disk-susceptibility testing. An MIC of ≥100 µg/mL was considered resistant for azlocillin and resistant to tobramycin if the MIC was ≥8 µg/mL.(*) |
| *Stack et al., 1998* | Susceptibility was assessed at baseline, and at 4 or 8 wk after treatment by E-test. Resistance was considered with bacterial growth at a drug concentration of >2 µg/mL for clarithromycin. (*) |
| *Walsh et al., 1993* | Susceptibility was assessed at baseline and for organisms culturable after the end of therapy and a 2 wk follow-up period by a microtiter tube dilution technique. Organisms were declared resistant if the MIC was greater than 2 µg/mL for rifampicin, greater than 8 µg/mL for novobiocin, and greater than 2 µg/mL and 38 µg/mL for trimethoprim and sulfamethoxazole. |
| *Winston et al., 1986* | Susceptibility of surveillance cultures was assessed at baseline, twice weekly during the study period and after study completion. Acquired organisms were defined as new organisms isolated during the study period, that were not present at baseline. An MIC ≤16 µg/mL was considered as sensitive for norfloxacin, polymyxin. For disc sensitivity testing cultures were considered sensitive to norfloxacin if a zone of ≥17 mm was present in a 10 µg norfloxacin disk. (*) |
| *Winston et al., 1990* | New organisms that were isolated during the study period but had not been present before the study were defined as acquired organisms. Susceptibility tests were done by agar dilution method, or by antibiotic disks. An MIC of ≤4, 16, or 4 µg/mL for ofloxacin, polymyxin, or vancomycin was considered susceptible to the antibiotics, respectively. For ofloxacin additional disk sensitivity testing was performed. Susceptibility was declared if a zone of 16 mm or greater was present around a 5 µg disk of ofloxacin. (*) |
| *Wurzer et al., 1997* | Susceptibility was assessed pre-treatment and between 4 and 6 wk of follow-up by agar dilution, and micro broth dilution. An MIC concentration of ≤2 µg/mL indicated susceptibility for clarithromycin, and an MIC above 2 µg/mL resistance. An MIC lower or equal to 0.125 µg/mL for amoxycilin was considered susceptible and classified resistant if above 0.125 µg/mL. (*) |

## Appendix 4

### Sensitivity analysis for main estimates

To test the robustness of our main analyses, we performed sensitivity analyses based on the model choice and the risk of bias.

### 1. Model choice

For our analyses, we applied the random effects model 4 described in *Jackson et al., 2018* using the R package *metafor* (*Viechtbauer, 2010*). To test the robustness of our estimates to the model choice we reran the main analyses with the conventional random effects model (model 1 in *Jackson et al., 2018*) and a corresponding Bayesian version of model 4 in *Jackson et al., 2018*. For the sensitivity analyses the R packages *metafor* (*Viechtbauer, 2010*), and MetaStan (*Günhan et al., 2020*) with default settings were used. We observe that our estimates are typically robust to model choice (*Appendix 4—figures 1 and 2*). Only for *P. aeruginousa* our estimate was not robust in our sensitivity analysis, where the alternative two approaches showed no harm or benefit of using a higher number of antibiotics (*Appendix 4—figure 1*).

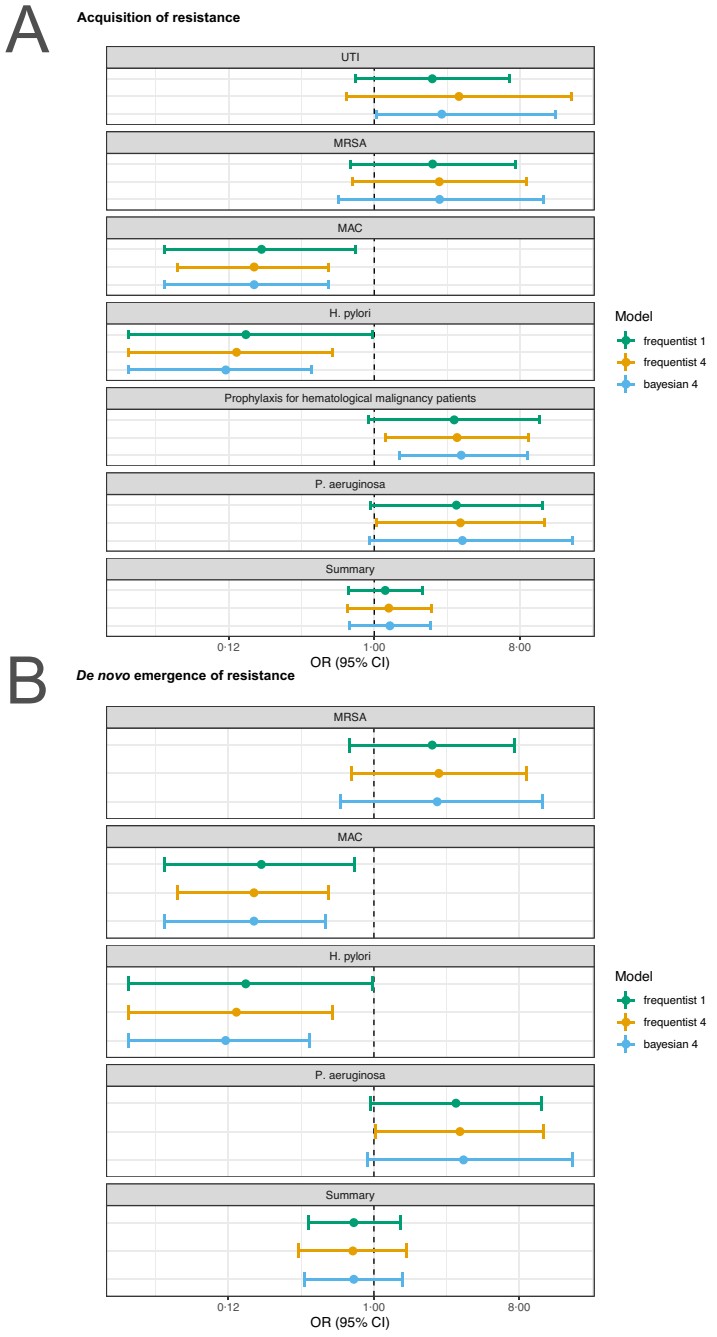

**Appendix 4—figure 1.** Sensitivity analysis based on model choice for the two main outcomes. (**A**) Acquisition of resistance, (**B**) de novo emergence of resistance. Shown are the frequentist model estimates of model 1, and model 4 presented in *Jackson et al., 2018* and a Bayesian estimate of model 4. UTI stands for urinary tract infection, MRSA for methicillin-resistant *Staphylococcus aureus*, and MAC for *Mycobacterium avium* complex.

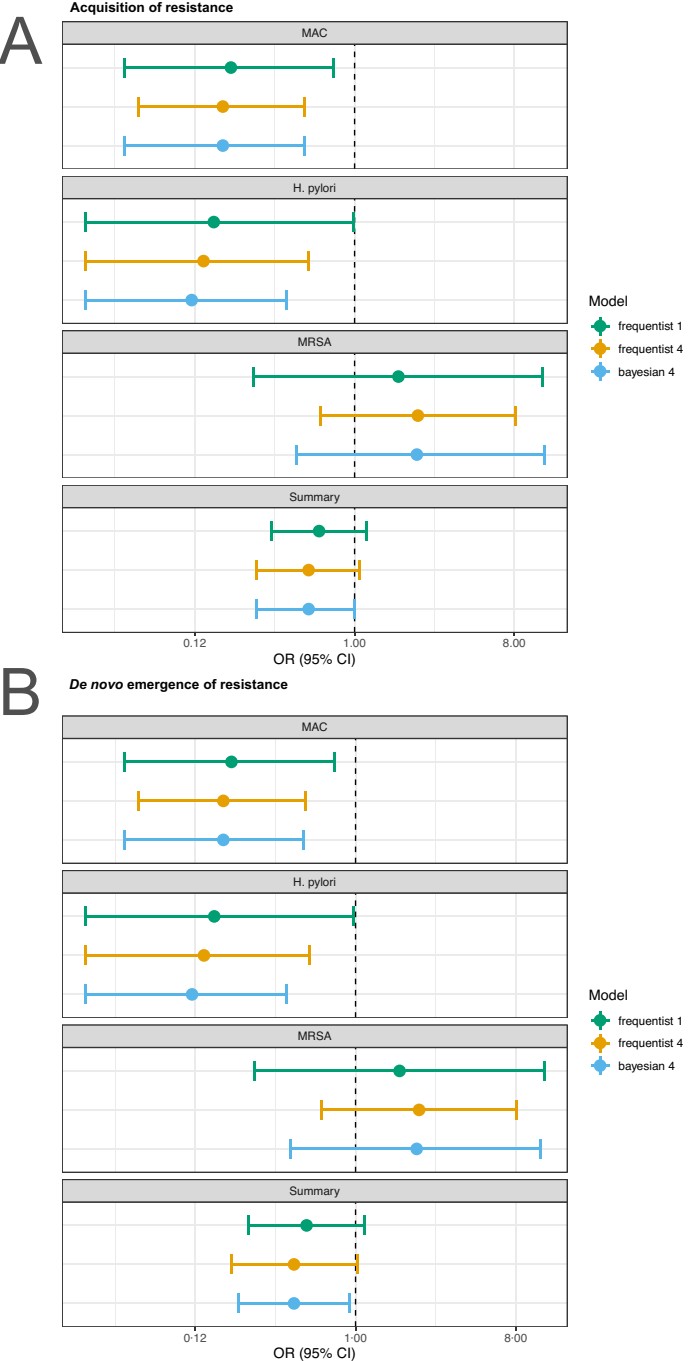

**Appendix 4—figure 2.** Sensitivity analysis based on model choice for the two main outcomes restricted to studies with at least one common antibiotic in the comparator arms. (**A**) Acquisition of resistance, (**B**) de novo emergence of resistance. Shown are the frequentist model estimates of model 1, and model 4 presented in *Jackson et al., 2018* and a Bayesian estimate of model 4.

## 2. Impact of risk of bias

To assess the impact of the risk of bias on our estimates, we reran the main analyses stratifying according to the overall risk of bias. For studies classified with an overall high risk of bias, our analysis shows that for acquisition of resistance using a lower number of antibiotics shows a benefit (pooled OR 4.45, 95% CI 1.67–11.81; $I^2$=57, *Appendix 4—table 1*). We did not observe any difference in using a higher number of antibiotics in comparison to less in resistance development when grouping the rest of the studies according to their risk assessment (*Appendix 4—table 1*). Nevertheless,

with less risk of bias administering a higher number of antibiotics seemed to perform better in comparison to less. However, no clear benefit could be determined (*Appendix 4—table 1*). This observation additionally supports that RCTs with resistance development as a main objective, and therefore potentially decreasing the risk of bias, are needed to understand the impact of different treatment strategies on antibiotic resistance outcomes.

**Appendix 4—table 1.** Summary of the results of the sub-group analyses stratifying according to the overall risk of bias for the two main outcomes.

Note that the listing of eligible studies also includes studies reporting zero cases in both treatment arms and were, therefore, not included in the statistical analysis.

| Overall risk of bias | Outcome | OR (95% CI) | Study heterogeneity ($I^2;\tau^2$) | Eligible studies |
|---|---|---|---|---|
| Some concerns | Acquisition of resistance | 0.71 (0.38–1.32) | 72%; 1.15 | *Bender et al., 1979; Black et al., 1982; Cometta et al., 1994; Dawson et al., 2015; Dekker et al., 1987; Dickstein et al., 2020; Dubé et al., 1997; Fournier et al., 1999; Gerecht et al., 1989; Harbarth et al., 2015; Hodson et al., 1987; Hultén et al., 1997; Iravani et al., 1981; Jo et al., 2021; Markowitz et al., 1992; Mavromanolakis et al., 1997; May et al., 1997; McCarty et al., 1988; Menon et al., 1986; Miehlke et al., 1998; Parras et al., 1995; Parry et al., 2007; Parry et al., 1977; Rubinstein et al., 1995; Schaeffer and Sisney, 1985; Stack et al., 1998; Walsh et al., 1993; Wurzer et al., 1997* |
| Some concerns | De novo emergence of resistance | 0.49 (0.21–1.14) | 73%; 1.53 | *Bender et al., 1979; Black et al., 1982; Cometta et al., 1994; Dawson et al., 2015; Dickstein et al., 2020; Dubé et al., 1997; Fournier et al., 1999; Gerecht et al., 1989; Harbarth et al., 2015; Hodson et al., 1987; Hultén et al., 1997; Iravani et al., 1981; Jo et al., 2021; Markowitz et al., 1992; May et al., 1997; McCarty et al., 1988; Miehlke et al., 1998; Parras et al., 1995; Parry et al., 1977; Rubinstein et al., 1995; Stack et al., 1998; Walsh et al., 1993; Wurzer et al., 1997* |
| High | Acquisition of resistance | 4.45 (1.67–11.81) | 57%; 1.11 | *Chaisson et al., 1997; Gibson et al., 1989; Haase et al., 1984; Hoepelman et al., 1988; Jacobs et al., 1993; Macnab et al., 1994; Paul et al., 2015; Pogue et al., 2021; Schaeffer et al., 1981; Smith et al., 1999; Winston et al., 1990; Winston et al., 1986* |
| High | De novo emergence of resistance | 2.32 (0.65–8.28) | 60%; 1.28 | *Chaisson et al., 1997; Gibson et al., 1989; Haase et al., 1984; Hoepelman et al., 1988; Jacobs et al., 1993; Macnab et al., 1994; Paul et al., 2015; Pogue et al., 2021; Smith et al., 1999* |

## Appendix 5

### Publication bias

In the study protocol, we stated that we will test for publication bias via visual inspection of the funnel plots and by Egger's test. As Egger's test can have problems with false-positive results for dichotomous outcomes, we used a modified version of the Egger's test, i.e., the Harbord's test (*Harbord et al., 2006*).

Neither the visual inspection of the funnel plots (*Appendix 5—figure 1*), nor Harbord's tests gave an indication of a publication bias for our two main outcomes acquisition, and de novo emergence of resistance (acquisition of resistance: Harbord's: p=0.28; de novo emergence of resistance: Harbord's: p=0.51).

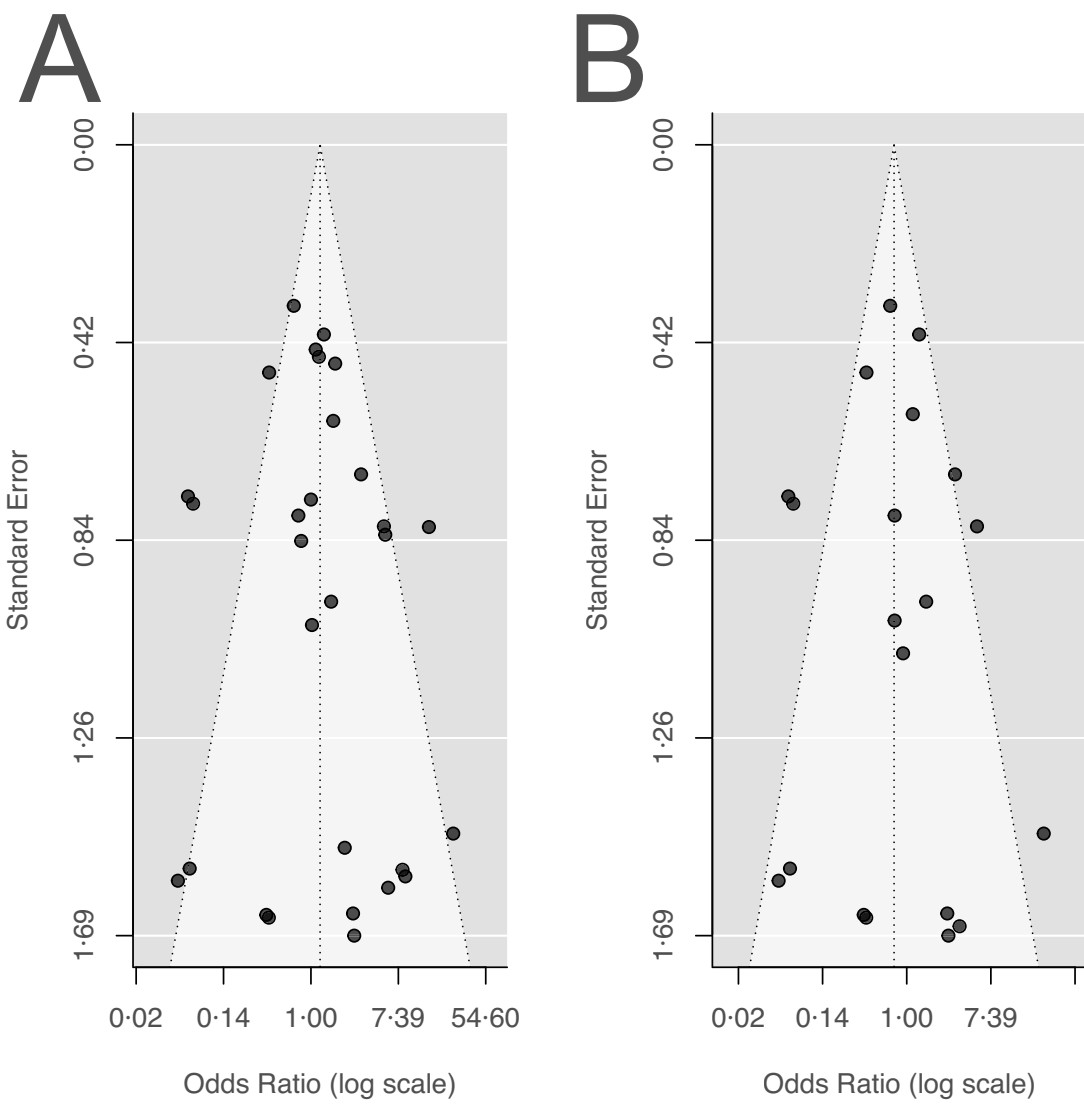

**Appendix 5—figure 1.** Funnel plots for the two main outcomes. (**A**) Acquisition of resistance, (**B**) de novo emergence of resistance.

# Appendix 6

## Sub-group analyses

The performance of an antibiotic treatment strategy to minimise resistance spread is not only dependent on the number of antibiotics administered. In our main estimates, we found a substantial amount of heterogeneity (main text: *Figure 3*; *Appendix 2—figure 1*), which is an indication that additional factors might be important to consider in a statistical model. In the following we first present the results of the pre-defined subgroup analyses from the study protocol and afterwards additional post-hoc subgroup analyses. One must consider that the results are mainly based on underpowered studies (main text: *Figure 2B*), and that in the subgroup analyses the number of included studies decreases. Therefore, the results of the subgroup analyses should be considered with care.

## 1. Predefined in the study protocol

The results of our subgroup-analyses for the outcome acquisition of resistance and de novo emergence of resistance are summarised in *Appendix 6—table 1* and table *Appendix 6—table 2*, respectively. The rationale for carrying out the predefined sub-group analyses are explained in the following subsections.

**Appendix 6—table 1.** Summary of the results of the predefined sub-group analyses for the outcome acquisition of resistance.

Note that the listing of eligible studies also includes studies reporting zero cases in both treatment arms, which are not included in the statistical analysis.

| Sub-group Analysis | OR (95% CI) | Study heterogeneity ($I^2$; $\tau^2$) | Eligible studies |
|---|---|---|---|
| **Number of antibiotics administered:** | | | |
| 1 vs 2 | 1.49 (0.77–2.88) | 76%; 1.70 | *Bender et al., 1979; Black et al., 1982; Cometta et al., 1994; Dickstein et al., 2020; Durante-Mangoni et al., 2013; Gerecht et al., 1989; Gibson et al., 1989; Haase et al., 1984; Hodson et al., 1987; Hoepelman et al., 1988; Hultén et al., 1997; Iravani et al., 1981; Jacobs et al., 1993; Jo et al., 2021; Markowitz et al., 1992; Mavromanolakis et al., 1997; McCarty et al., 1988; Menon et al., 1986; Miehlke et al., 1998; Parry et al., 2007; Parry et al., 1977; Paul et al., 2015; Pogue et al., 2021; Pujol et al., 2021; Rubinstein et al., 1995; Schaeffer et al., 1981; Schaeffer and Sisney, 1985; Smith et al., 1999; Stack et al., 1998; Winston et al., 1990; Winston et al., 1986; Wurzer et al., 1997* |
| 2 vs 3 | 0.38 (0.08–1.78) | 74%; 1.63 | *Chaisson et al., 1997; Dubé et al., 1997; Fournier et al., 1999; May et al., 1997; Walsh et al., 1993* |
| **Administration of additional non-antibiotic drugs:** | | | |
| Non-antibiotic drugs as part of treatment | 0.88 (0.21–3.66) | 82%; 3.00 | *Bender et al., 1979; Dekker et al., 1987; Hultén et al., 1997; Miehlke et al., 1998; Parras et al., 1995; Stack et al., 1998; Winston et al., 1990; Winston et al., 1986; Wurzer et al., 1997* |
| Non-antibiotic drugs administered if necessary | 1.07 (0.48–2.40) | 1%; 0.01 | *Dickstein et al., 2020; Durante-Mangoni et al., 2013; Iravani et al., 1981; Jacobs et al., 1993; Pujol et al., 2021* |
| Usage of the same dosage of antibiotics common to both treatment arms | 0.59 (0.30–1.18) | 73%; 1.20 | *Bender et al., 1979; Chaisson et al., 1997; Cometta et al., 1994; Dickstein et al., 2020; Dubé et al., 1997; Durante-Mangoni et al., 2013; Fournier et al., 1999; Hultén et al., 1997; Jacobs et al., 1993; May et al., 1997; McCarty et al., 1988; Parry et al., 1977; Pogue et al., 2021; Pujol et al., 2021; Smith et al., 1999; Stack et al., 1998; Walsh et al., 1993; Wurzer et al., 1997* |
| **Required comorbidity at study inclusion:** | | | |
| Yes | 1.23 (0.50–3.01) | 72%; 1.59 | *Bender et al., 1979; Chaisson et al., 1997; Dekker et al., 1987; Dubé et al., 1997; Fournier et al., 1999; Gibson et al., 1989; Hodson et al., 1987; Jacobs et al., 1993; Markowitz et al., 1992; May et al., 1997; McCarty et al., 1988; Parry et al., 1977; Smith et al., 1999; Winston et al., 1990; Winston et al., 1986* |

*Appendix 6—table 1 Continued on next page*

*Appendix 6—table 1 Continued*

| Sub-group Analysis | OR (95% CI) | Study heterogeneity ($I^2$; $\tau^2$) | Eligible studies |
|---|---|---|---|
| No | 1.25 (0.55–2.86) | 80%; 2.02 | *Black et al., 1982; Cometta et al., 1994; Dawson et al., 2015; Dickstein et al., 2020; Durante-Mangoni et al., 2013; Gerecht et al., 1989; Haase et al., 1984; Harbarth et al., 2015; Hoepelman et al., 1988; Hultén et al., 1997; Iravani et al., 1981; Jo et al., 2021; Macnab et al., 1994; Mavromanolakis et al., 1997; Menon et al., 1986; Miehlke et al., 1998; Parras et al., 1995; Parry et al., 2007; Paul et al., 2015; Pogue et al., 2021; Pujol et al., 2021; Rubinstein et al., 1995; Schaeffer et al., 1981; Schaeffer and Sisney, 1985; Stack et al., 1998; Walsh et al., 1993; Wurzer et al., 1997* |
| Gram-status | | | |
| Negative | 1.14 (0.56–2.35) | 78%; 1.57 | *Black et al., 1982; Dekker et al., 1987; Dickstein et al., 2020; Durante-Mangoni et al., 2013; Hodson et al., 1987; Hultén et al., 1997; Iravani et al., 1981; Jo et al., 2021; Mavromanolakis et al., 1997; McCarty et al., 1988; Menon et al., 1986; Miehlke et al., 1998; Parry et al., 2007; Parry et al., 1977; Pogue et al., 2021; Rubinstein et al., 1995; Schaeffer and Sisney, 1985; Smith et al., 1999; Stack et al., 1998; Winston et al., 1990; Winston et al., 1986; Wurzer et al., 1997* |
| Positive | 0.44 (0.11–1.76) | 66%; 1.54 | *Chaisson et al., 1997; Dubé et al., 1997; Fournier et al., 1999; Harbarth et al., 2015; Markowitz et al., 1992; May et al., 1997; Parras et al., 1995; Paul et al., 2015; Pujol et al., 2021; Walsh et al., 1993* |
| Negative and positive | 3.38 (1.08–10.58) | 44%; 0.75 | *Bender et al., 1979; Cometta et al., 1994; Gerecht et al., 1989; Gibson et al., 1989; Haase et al., 1984; Hoepelman et al., 1988; Jacobs et al., 1993; Schaeffer et al., 1981* |
| Only resistances of antibiotics common to treatment arms | 0.39 (0.18–0.81) | 75%; 1.49 | *Bender et al., 1979; Black et al., 1982; Chaisson et al., 1997; Cometta et al., 1994; Dawson et al., 2015; Dickstein et al., 2020; Dubé et al., 1997; Durante-Mangoni et al., 2013; Fournier et al., 1999; Hultén et al., 1997; Jacobs et al., 1993; Macnab et al., 1994; May et al., 1997; McCarty et al., 1988; Menon et al., 1986; Parry et al., 2007; Parry et al., 1977; Pogue et al., 2021; Pujol et al., 2021; Smith et al., 1999; Stack et al., 1998; Walsh et al., 1993; Wurzer et al., 1997* |
| Age of antibiotics since the conduction of the trial: | | | |
| Youngest antibiotic is in the treatment arm with the lower number of antibiotics | 1.63 (0.66–4.03) | 76%; 2.17 | *Black et al., 1982; Dawson et al., 2015; Dekker et al., 1987; Dubé et al., 1997; Gerecht et al., 1989; Gibson et al., 1989; Haase et al., 1984; Harbarth et al., 2015; Hodson et al., 1987; Hoepelman et al., 1988; Hultén et al., 1997; Jacobs et al., 1993; McCarty et al., 1988; Menon et al., 1986; Parras et al., 1995; Pujol et al., 2021; Schaeffer and Sisney, 1985; Smith et al., 1999; Stack et al., 1998; Winston et al., 1990; Winston et al., 1986* |
| Youngest antibiotic is in the treatment arm with the higher number of antibiotics | 1.08 (0.49–2.42) | 66%; 0.91 | *Chaisson et al., 1997; Dickstein et al., 2020; Durante-Mangoni et al., 2013; Fournier et al., 1999; Iravani et al., 1981; Jo et al., 2021; Markowitz et al., 1992; Mavromanolakis et al., 1997; May et al., 1997; Miehlke et al., 1998; Parry et al., 2007; Parry et al., 1977; Paul et al., 2015; Pogue et al., 2021; Rubinstein et al., 1995; Schaeffer et al., 1981; Walsh et al., 1993* |
| No antibiotics common to treatment arms | 4.73 (2.14–10.42) | 37%; 0.51 | *Dekker et al., 1987; Gerecht et al., 1989; Gibson et al., 1989; Haase et al., 1984; Harbarth et al., 2015; Hodson et al., 1987; Hoepelman et al., 1988; Iravani et al., 1981; Jo et al., 2021; Markowitz et al., 1992; Mavromanolakis et al., 1997; Miehlke et al., 1998; Parras et al., 1995; Paul et al., 2015; Rubinstein et al., 1995; Schaeffer et al., 1981; Schaeffer and Sisney, 1985; Winston et al., 1990; Winston et al., 1986* |

**Appendix 6—table 2.** Summary of the results of the predefined sub-group analyses for the outcome de novo emergence of resistance.

Note that the listing of eligible studies also includes studies reporting zero cases of resistance in both treatment arms, which were, therefore, not included in the statistical analysis.

| Sub-group analysis | OR (95% CI) | Study heterogeneity ($I^2$; $\tau^2$) | Eligible studies |
|---|---|---|---|
| Number of antibiotics administered: | | | |
| 1 vs 2 | 0.89 (0.38–2.11) | 75%; 1.90 | *Bender et al., 1979; Black et al., 1982; Cometta et al., 1994; Dickstein et al., 2020; Durante-Mangoni et al., 2013; Gerecht et al., 1989; Gibson et al., 1989; Hodson et al., 1987; Hoepelman et al., 1988; Hultén et al., 1997; Iravani et al., 1981; Jacobs et al., 1993; Jo et al., 2021; Markowitz et al., 1992; McCarty et al., 1988; Miehlke et al., 1998; Parry et al., 1977; Paul et al., 2015; Pogue et al., 2021; Pujol et al., 2021; Rubinstein et al., 1995; Smith et al., 1999; Stack et al., 1998; Wurzer et al., 1997* |

*Appendix 6—table 2 Continued on next page*

*Appendix 6—table 2 Continued*

| Sub-group analysis | OR (95% CI) | Study heterogeneity ($I^2$; $\tau^2$) | Eligible studies |
|---|---|---|---|
| 2 vs 3 | 0.38 (0.08–1.78) | 74%; 1.63 | *Chaisson et al., 1997*; *Dubé et al., 1997*; *Fournier et al., 1999*; *May et al., 1997*; *Walsh et al., 1993* |
| **Administration of additional non-antibiotic drugs:** | | | |
| Non-antibiotic drugs as part of treatment | 0.22 (0.04–1.10) | 82%; 1.10 | *Bender et al., 1979*; *Hultén et al., 1997*; *Miehlke et al., 1998*; *Parras et al., 1995*; *Stack et al., 1998*; *Wurzer et al., 1997* |
| Non-antibiotic drugs administered if necessary | 0.97 (0.36–2.58) | 1%; 0.01 | *Dickstein et al., 2020*; *Durante-Mangoni et al., 2013*; *Iravani et al., 1981*; *Jacobs et al., 1993*; *Pujol et al., 2021* |
| Usage of the same dosage of antibiotics common to both treatment arms | 0.53 (0.24–1.16) | 71%; 1.38 | *Bender et al., 1979*; *Chaisson et al., 1997*; *Cometta et al., 1994*; *Dickstein et al., 2020*; *Dubé et al., 1997*; *Durante-Mangoni et al., 2013*; *Fournier et al., 1999*; *Hultén et al., 1997*; *Jacobs et al., 1993*; *May et al., 1997*; *McCarty et al., 1988*; *Parry et al., 1977*; *Pogue et al., 2021*; *Pujol et al., 2021*; *Smith et al., 1999*; *Stack et al., 1998*; *Walsh et al., 1993*; *Wurzer et al., 1997* |
| **Required comorbidity at study inclusion:** | | | |
| Yes | 0.71 (0.21–2.41) | 67%; 1.57 | *Bender et al., 1979*; *Black et al., 1982*; *Chaisson et al., 1997*; *Cometta et al., 1994*; *Dawson et al., 2015*; *Dickstein et al., 2020*; *Dubé et al., 1997*; *Durante-Mangoni et al., 2013*; *Fournier et al., 1999*; *Hultén et al., 1997*; *Jacobs et al., 1993*; *Macnab et al., 1994*; *May et al., 1997*; *McCarty et al., 1988*; *Parry et al., 1977*; *Pogue et al., 2021*; *Pujol et al., 2021*; *Smith et al., 1999*; *Stack et al., 1998*; *Walsh et al., 1993*; *Wurzer et al., 1997* |
| No | 0.75 (0.28–2.01) | 80%; 2.02 | *Gerecht et al., 1989*; *Gibson et al., 1989*; *Harbarth et al., 2015*; *Hodson et al., 1987*; *Hoepelman et al., 1988*; *Iravani et al., 1981*; *Jo et al., 2021*; *Markowitz et al., 1992*; *Miehlke et al., 1998*; *Parras et al., 1995*; *Paul et al., 2015*; *Rubinstein et al., 1995* |
| **Gram status** | | | |
| Negative | 0.60 (0.23–1.55) | 78%; 1.59 | *Black et al., 1982*; *Dickstein et al., 2020*; *Durante-Mangoni et al., 2013*; *Hodson et al., 1987*; *Hultén et al., 1997*; *Iravani et al., 1981*; *Jo et al., 2021*; *McCarty et al., 1988*; *Miehlke et al., 1998*; *Parry et al., 1977*; *Pogue et al., 2021*; *Rubinstein et al., 1995*; *Smith et al., 1999*; *Stack et al., 1998*; *Wurzer et al., 1997* |
| Positive | 0.44 (0.11–1.76) | 66%; 1.54 | *Chaisson et al., 1997*; *Dubé et al., 1997*; *Fournier et al., 1999*; *Harbarth et al., 2015*; *Markowitz et al., 1992*; *May et al., 1997*; *Parras et al., 1995*; *Paul et al., 2015*; *Pujol et al., 2021*; *Walsh et al., 1993* |
| Negative and positive | 3.34 (0.59–18.97) | 47%; 1.39 | *Bender et al., 1979*; *Cometta et al., 1994*; *Gerecht et al., 1989*; *Gibson et al., 1989*; *Hoepelman et al., 1988*; *Jacobs et al., 1993* |
| Only resistances of antibiotics common to treatment arms | 0.32 (0.16–0.66) | 59%; 0.87 | *Bender et al., 1979*; *Black et al., 1982*; *Chaisson et al., 1997*; *Cometta et al., 1994*; *Dawson et al., 2015*; *Dickstein et al., 2020*; *Dubé et al., 1997*; *Durante-Mangoni et al., 2013*; *Fournier et al., 1999*; *Hultén et al., 1997*; *Jacobs et al., 1993*; *Macnab et al., 1994*; *May et al., 1997*; *McCarty et al., 1988*; *Parry et al., 1977*; *Pogue et al., 2021*; *Pujol et al., 2021*; *Smith et al., 1999*; *Stack et al., 1998*; *Walsh et al., 1993*; *Wurzer et al., 1997* |
| **Age of antibiotics since the conduction of the trial:** | | | |
| Youngest antibiotic is in the treatment arm with the lower number of antibiotics | 0.73 (0.19–2.77) | 75%; 2.83 | *Black et al., 1982*; *Dawson et al., 2015*; *Dubé et al., 1997*; *Gerecht et al., 1989*; *Gibson et al., 1989*; *Harbarth et al., 2015*; *Hodson et al., 1987*; *Hoepelman et al., 1988*; *Hultén et al., 1997*; *Jacobs et al., 1993*; *McCarty et al., 1988*; *Parras et al., 1995*; *Pujol et al., 2021*; *Smith et al., 1999*; *Stack et al., 1998* |
| Youngest antibiotic is in the treatment arm with the higher number of antibiotics | 0.86 (0.34–2.17) | 70%; 0.98 | *Chaisson et al., 1997*; *Dickstein et al., 2020*; *Durante-Mangoni et al., 2013*; *Fournier et al., 1999*; *Iravani et al., 1981*; *Jo et al., 2021*; *Markowitz et al., 1992*; *May et al., 1997*; *Miehlke et al., 1998*; *Parry et al., 1977*; *Paul et al., 2015*; *Pogue et al., 2021*; *Rubinstein et al., 1995*; *Walsh et al., 1993* |
| No antibiotics common to treatment arms | 3.54 (0.91–13.75) | 38%; 0.68 | *Gerecht et al., 1989*; *Gibson et al., 1989*; *Harbarth et al., 2015*; *Hodson et al., 1987*; *Hoepelman et al., 1988*; *Iravani et al., 1981*; *Jo et al., 2021*; *Markowitz et al., 1992*; *Miehlke et al., 1998*; *Parras et al., 1995*; *Paul et al., 2015*; *Rubinstein et al., 1995* |

## i. Number of antibiotics administered

In our systematic review, we did not predefine a fixed number of antibiotics to compare. We rather aimed to investigate whether there is a general trend of a treatment strategy with a higher number of antibiotics performing better than one with fewer antibiotics with respect to resistance

development. One can imagine though, that the magnitude of this trend might vary depending on the number of antibiotics compared. For example, if resistance against the used antibiotics is likely to be encountered in the population, a comparison of one versus two antibiotics might give different results than two versus three. In the 1960s for Mtb, the number of antibiotics was for instance increased to three antibiotics at the initial treatment phase, due to the finding that primary resistance can be encountered for one drug but rarely to two or three antibiotics (*Fox et al., 1999*; *Fox et al., 1957*; *Mitchison and Selkon, 1957*). On one hand, if the number of antibiotics used is rather high in both treatment arms, there might be no difference in resistance development detected as the treatment period might be too short to observe a relevant effect. On the other hand, if the treatment period is rather long, there might also not be an efficient effect detectable when a low number of antibiotics is compared and the timespan between follow-up cultures is long. We considered the effect of treatment length, and length of follow-up on our estimates later in the meta-regression and multi-model inference (Appendix 7).

We identified three studies comparing one versus three antibiotics, but two of them had zero events for both comparator arms. We included two Mtb studies in our review comparing three versus four antibiotics, but both had zero events in the comparator arms. For the estimates of one versus two antibiotics and two versus three antibiotics we did not identify a difference in using a higher number of antibiotics, in comparison to less, and substantial heterogeneity was observed (*Appendix 6—table 1*, *Appendix 6—table 2*). Nevertheless, for the estimate of two versus three antibiotics, there was a beneficial trend for using a higher number of antibiotics observable. However, no clear benefit could be determined (*Appendix 6—table 1*, *Appendix 6—table 2*). This might indicate that in general a higher number of antibiotics in treatments is beneficial.

## ii. Administration of additional non-antibiotic drugs

In our inclusion criteria, we allowed the administration of additional non-antibiotic drugs, which potentially could also affect the resistance outcome due to faster cure of patients, or by specifically supporting the activity of antibiotics, such as e.g., beta-lactam-inhibitors. To test the effect of the administration of additional non-administered antibiotics, we performed a sub-group analysis based on whether a study administered additional non-antibiotic drugs or not. Notably, in our studies, the additional non-antibiotics were always administered in both treatment arms. Considering if additional non-antibiotic drugs were administered or not did not show any harm or benefit on the resistance outcome whether a higher number of antibiotics was used or a lower number (*Appendix 6—table 1*, *Appendix 6—table 2*). A few studies allowed the administration of additional non-antibiotics, but they were not a fixed part of the treatment regime. Also, in those studies, we did not identify a harm or benefit (*Appendix 6—table 1*, *Appendix 6—table 2*).

## iii. Usage of same dosage of antibiotics common to both treatment arms

Not only the number of total antibiotics might determine the efficacy of a treatment, but also the dosage of antibiotics. To compare more similar treatments, we estimated the pooled OR for studies that administered at least one antibiotic common to both treatment arms, and where additionally the antibiotics that were common were administered with the same dosage. We observed that in most cases if at least one common antibiotic was administered, their dosage was the same (78% acquisition of resistance, 86% emergence of resistance). Therefore, it is not surprising that we observe, in line with the analysis of 'at least one antibiotic common to both treatment arms' (main text: *Figure 3B*, *Appendix 2—figure 1B*), no difference in using a higher number of antibiotics in comparison to less to reduce resistances (*Appendix 6—table 1*, *Appendix 6—table 2*). In both cases, we observed a substantial amount of heterogeneity, which indicates that further factors might play a role for explaining the observed resistance differences.

## iv. Required comorbidity at study inclusion

The way the immune-system reacts to an infection might potentially influence the frequencies of resistances observed (*Handel et al., 2009*). Therefore, we tested whether studies that considered patients with a comorbidity, assuming that the immune system is to some extent compromised, show a different trend of resistance development in comparison to studies where no comorbidity was required for study inclusion. For this analysis, we considered studies, that had comorbidities as a requirement for study entry. We could not identify a difference in using a higher number of antibiotics in comparison to less for both main outcomes, regardless of comorbidity status at the study entrance (*Appendix 6—table 1*, *Appendix 6—table 2*).

### v. Study was conducted in an ICU

Another way to test the potential role of the immune system is by the severity of illness, approximated by whether the study population was treated within an ICU or not. We were not able to link on a patient level the data of resistance development to the patient's ICU status. Therefore, we tried to classify the ICU status per study, i.e., one status for the whole study population. We only identified two studies (5%) for the acquisition of resistance, where the whole study population was in the ICU. We found 9 studies (21%), where no patient was treated in the ICU. For the rest of the studies the population could either be mixed (14%), or no information was confidentially extractable (60%). Since the ICU status on a study level seemed to be an uninformative proxy, we decided not to perform sub-group analyses for this factor.

### vi. Gram-status

The gram status of a bacterium may potentially determine how effective an antibiotic, or an antibiotic combination is. Differences between gram-negative and gram-positive bacteria such as distinct bacterial surface organisation can lead to specific intrinsic resistances of gram-negative and gram-positive bacteria against antibiotics (*Exner et al., 2017*). These structural differences can lead to varying effects of antibiotic combinations between gram-negative and gram-positive bacteria (*Cacace et al., 2023*). Additionally, plasmids play a major role in the dissemination of antibiotic resistance genes in both gram-positive, and negative bacteria (*Vrancianu et al., 2020*). The spread of plasmids differs considerably between gram-positive bacteria and gram-negative bacteria (*Goessweiner-Mohr et al., 2014*). These structural differences could influence the performance of antibiotic treatment strategies. To test the influence of the gram-status on our estimates we performed sub-group analyses with studies, that focused only on measurements of gram-negative bacteria, gram-positive, or both. We classified the gram-status on a study level as we could not link the gram-status and resistance development on a patient level.

When selecting for studies that either focus on gram-negative, or gram-positive we did not identify a difference in using a higher number of antibiotics in comparison to less for both main outcomes (*Appendix 6—table 1*, *Appendix 6—table 2*). For the subgroup analysis including studies with a focus on both gram-negative and positive bacteria, the treatment strategy with a lower number of antibiotics showed a benefit for the main outcome acquisition of resistance (pooled OR 3.38, 95% CI 1.08–10.58; $I^2$=44%, *Appendix 6—table 1*). However, for de novo emergence of resistance we did not identify a difference (pooled OR 3.35, 95% CI 0.67–16.71; $I^2$=47%, *Appendix 6—table 2*) It seems, that acquisition of resistance is more sensible to the restriction on which gram-status is considered. This might be due to the broader definition of acquisition of resistance as it is more sensitive to resistance changes in the microbial community. If a treatment is targeted against a specific pathogen, e.g., a gram-positive bacterium, other bacteria of the microbiota are exposed to the treatment as well. Some bacteria of the microbiota might be more intrinsically resistant against the administered antibiotics, e.g., a gram-negative bacterium, and are, therefore, more likely to develop resistance. With the acquisition of resistance, we might detect such effects.

### vii. All resistances not only against administered antibiotics

Antibiotic resistances can be acquired by plasmids, which in a clinical context often confer resistances against multiple antibiotics (*Cazares et al., 2020*; *Holt et al., 2011*; *Paterson and Bonomo, 2005*). Therefore, we aimed to test, whether a higher number of antibiotics also leads to resistance against a higher number of antibiotics, considering both resistances to antibiotics that were administered and ones that were not. For the acquisition of resistance, we only identified seven studies that measured resistances also against non-administered drugs. Only three of those studies have non-zero events. For de novo emergence of resistance, we identified four studies measuring resistances against non-administered antibiotics, were two of them have non-zero events in both treatment arms. Due to the small number of studies identified and an even smaller number of studies having non-zero resistance events, we only present the estimates of the resistances against non-administered antibiotics (Appendix 9 sections xi. and xii.).

### viii. Only resistances of antibiotics common to treatments arms

To estimate how the same antibiotics performed in the different treatment arms we performed a subgroup-analysis only considering resistance against antibiotics common to both treatment arms. For both main outcomes, we observed that if we only consider resistances of common antibiotics the treatment arm with the higher number of antibiotics showed a benefit (acquisition of resistance:

pooled OR 0.39, 95% CI 0.18–0.81; $I^2$=75%, **Appendix 6—table 1**; emergence of resistance: pooled OR 0.32, 95% CI 0.16–0.66; $I^2$=87% **Appendix 6—table 2**). Consequently, we can conclude that for a specific antibiotic less resistances will develop in a treatment arm with a higher number of antibiotics.

As the studies included in our meta-analysis often did not quantify the resistance outcome for all antibiotics administered in a treatment arm it is harder to assess the full resistance burden of the antibiotic treatments systematically. One could argue that due to the higher number of antibiotics given in one treatment arm, one would also observe in total a higher resistance burden in that arm. This possible effect could be magnified dependent on the potency of antibiotics. If a treatment arm is a combination of low potency antibiotics, one might expect a higher chance of resistance. The results of this sub-group analysis highlight once more that a systematic exploration of resistance development in RCTs is important for a better understanding of resistance development during treatment and that the identity of the administered antibiotics might play an important role.

## ix. Age of antibiotics since the conduction of the trial

The prevalence of antibiotic resistance affects the treatment's success. If resistance before treatment is frequent in the population, then this increases the likelihood that the prescribed antibiotic treatment fails for any patient. We collected data on the year the admission of patients for the individual studies started and the year antibiotics became available. With the naive assumption that the longer the antibiotic had been available before the study was conducted, the higher is its resistance prevalence within the population. This assumption has its weaknesses as antibiotics are used with different intensities over the years and their local pattern of use might vary. However, such data are more difficult to retrieve. Hence, the years an antibiotic was available until the trial started is a simple first approximation to investigate resistance prevalence.

If the studies did not state the year the trial started, we extracted the publication year. For the availability of antibiotics, we used the older of the two dates available on DrugBank (**Wishart et al., 2018**) and DrugCentral (**Ursu et al., 2019**) (DrugBank: marketing start, DrugCentral: approvals).

In the following, we present the subgroup analyses, where we classify in which comparator arm the youngest antibiotic is administered. We did not detect a harm or benefit of using a higher or lower number of antibiotics when stratifying, and observed in all subgroup analyses at least a substantial amount of heterogeneity (**Appendix 6—table 1**, **Appendix 6—table 2**).

Furthermore, we performed subgroup analyses stratifying according to the mean age of antibiotics in a treatment arm, and the oldest antibiotic of the treatment arm. For those analyses, we also did not identify a difference of using a higher number of antibiotics over fewer. It could be that our approximation is too simplified to estimate the potential effect.

## x. No antibiotics common to treatment arms

In the main analyses, we presented the estimates for all studies, and studies, which administered at least one antibiotic common to the treatment arms. Here in the Appendix we present the resistance estimates for less comparable treatments, i.e. for studies, whose treatment arms had no antibiotics in common. For those studies, we observed for both main outcomes a trend favouring the treatment arm with fewer antibiotics (acquisition of resistance: pooled OR 4.73, 95% CI 2.14–10.42; $I^2$=37%, **Appendix 6—table 1**; de novo emergence of resistance: pooled OR 3.54, 95% CI 0.91–13.75; $I^2$=38%, **Appendix 6—table 2**). The benefit was for the acquisition of resistance clear, and for de novo emergence of resistance not. The result that if the treatment arms had no antibiotics in common a lower number of antibiotics performed better than a higher number of antibiotics could be due to different potencies of antibiotics or resistance prevalences. Further, there could be a bias to combine less potent antibiotics or antibiotics with higher resistance prevalence to ensure treatment efficacy, which could lead to higher chances of detecting resistances in the treatment arm with the higher number of antibiotics, e.g. by selecting pre-existing resistance (see also Appendix 6 section ix.). This highlights once more that the identity of antibiotics may play an important role in determining whether combining antibiotics is beneficial or not with respect to resistance development.

## xi. Systematic testing of the whole study population

In our protocol, we predefined that we would perform sub-group analyses based on whether the resistance data were systematically available for the whole study population or just a subset of patients. All our included studies attempted to measure resistance data for the whole study population. In some cases, more information on resistance development was reported than what we could use. In those cases, it was impossible to distinguish how many patients were evaluable for

the resistance outcomes, and/or how many patients developed resistances. In summary, we always obtained data for the whole study population, except for the missing data cases, but nevertheless, we could not process all information given due to the way it was reported. The influence of missing data is assessed in the risk of bias assessment (Appendix 3), and the corresponding sensitivity analyses (Appendix 4 section 2).

## 2. Post-hoc subgroup analyses
### i. Additional administration of antibiotics

During our selection process of studies, we realised that some studies allowed the addition of further antibiotics to the assigned treatments, if necessary, whereas others explicitly stated no other antibiotics than the assigned ones are given during the treatment phase. For a large proportion of all included studies, we could not extract whether additional antibiotics were allowed or not (62%). As we cannot rule out that in those studies no additional antibiotics were administered, we decided to include studies where additional antibiotics are allowed. To check the impact of this decision we performed sub-group analyses for those studies, where information on the administration of additional antibiotics was given. We identified 12 studies, which allowed the administration of additional antibiotics, but only at most seven studies could be included in the statistical analyses as the other trials reported zero cases in both treatment arms (*Durante-Mangoni et al., 2013*; *Gibson et al., 1989*; *Harbarth et al., 2015*; *Iravani et al., 1981*; *McCarty et al., 1988*, *Appendix 6—table 3*, *Appendix 6—table 4*). We identified three studies explicitly excluding additional antibiotics, however, the statistical analyses are based on two studies as one reported zero cases in both treatment arms (*Gerecht et al., 1989*;*Appendix 6—table 3*, *Appendix 6—table 4*). Therefore, the impact of allowing the administration of additional antibiotics, if necessary, on our overall estimates was difficult to infer.

**Appendix 6—table 3.** Summary of the results of the post-hoc sub-group analyses for the outcome acquisition of resistance.

Note that the listing of eligible studies also includes studies reporting zero cases in both treatment arms and were, therefore, not included in the statistical analysis.

| Sub-group Analysis | OR (95% CI) | Study heterogeneity ($I^2$; $\tau^2$) | Eligible studies |
|---|---|---|---|
| Additional administration of antibiotics: | | | |
| Allowed | 1.18 (0.70–1.97) | 16%; 0.07 | *Dickstein et al., 2020*; *Durante-Mangoni et al., 2013*; *Gibson et al., 1989*; *Harbarth et al., 2015*; *Iravani et al., 1981*; *Jacobs et al., 1993*; *McCarty et al., 1988*; *Paul et al., 2015*; *Pogue et al., 2021*; *Pujol et al., 2021*; *Rubinstein et al., 1995*; *Winston et al., 1990* |
| Prohibited | 0.19 (0.04–0.98) | 57%; 0.79 | *Gerecht et al., 1989*; *Hultén et al., 1997*; *Wurzer et al., 1997* |
| Pre-resistance against non-administered antibiotics required at study inclusion: | | | |
| Required | 1.08 (0.57–2.05) | 15%; 0.07 | *Dickstein et al., 2020*; *Durante-Mangoni et al., 2013*; *Harbarth et al., 2015*; *Parras et al., 1995*; *Paul et al., 2015*; *Pogue et al., 2021*; *Pujol et al., 2021*; *Walsh et al., 1993* |

*Appendix 6—table 3 Continued on next page*

*Appendix 6—table 3 Continued*

| Sub-group Analysis | OR (95% CI) | Study heterogeneity ($I^2$; $\tau^2$) | Eligible studies |
|---|---|---|---|
| No | 1.25 (0.61–2.55) | 79%; 2.22 | *Bender et al., 1979*; *Black et al., 1982*; *Chaisson et al., 1997*; *Cometta et al., 1994*; *Dawson et al., 2015*; *Dekker et al., 1987*; *Dubé et al., 1997*; *Fournier et al., 1999*; *Gerecht et al., 1989*; *Gibson et al., 1989*; *Haase et al., 1984*; *Hodson et al., 1987*; *Hoepelman et al., 1988*; *Hultén et al., 1997*; *Iravani et al., 1981*; *Jacobs et al., 1993*; *Jo et al., 2021*; *Macnab et al., 1994*; *Markowitz et al., 1992*; *Mavromanolakis et al., 1997*; *May et al., 1997*; *McCarty et al., 1988*; *Menon et al., 1986*; *Miehlke et al., 1998*; *Parry et al., 2007*; *Parry et al., 1977*; *Rubinstein et al., 1995*; *Schaeffer et al., 1981*; *Schaeffer and Sisney, 1985*; *Smith et al., 1999*; *Stack et al., 1998*; *Winston et al., 1990*; *Winston et al., 1986*; *Wurzer et al., 1997* |
| Way of antibiotic administration: | | | |
| Orally | 1.18 (0.44–3.15) | 78%; 2.70 | *Bender et al., 1979*; *Black et al., 1982*; *Chaisson et al., 1997*; *Dawson et al., 2015*; *Dubé et al., 1997*; *Haase et al., 1984*; *Hultén et al., 1997*; *Iravani et al., 1981*; *Jo et al., 2021*; *Macnab et al., 1994*; *Mavromanolakis et al., 1997*; *Menon et al., 1986*; *Miehlke et al., 1998*; *Parry et al., 2007*; *Schaeffer et al., 1981*; *Schaeffer and Sisney, 1985*; *Stack et al., 1998*; *Walsh et al., 1993*; *Winston et al., 1990*; *Winston et al., 1986*; *Wurzer et al., 1997* |
| Intravenously | 1.83 (0.67–5.00) | 66%; 0.90 | *Dickstein et al., 2020*; *Durante-Mangoni et al., 2013*; *Gerecht et al., 1989*; *Gibson et al., 1989*; *Hoepelman et al., 1988*; *Jacobs et al., 1993*; *Markowitz et al., 1992*; *Pogue et al., 2021*; *Pujol et al., 2021*; *Smith et al., 1999* |
| Different ways of administration in the treatment arms | 1.51 (0.67–3.39) | 1%; 0.01 | *Dekker et al., 1987*; *Hodson et al., 1987*; *Parry et al., 1977*; *Paul et al., 2015* |

**Appendix 6—table 4.** Summary of the results of the post-hoc sub-group analyses for the outcome de novo emergence of resistance.
Note that the listing of eligible studies also includes studies reporting zero cases in both treatment arms and were, therefore, not included in the statistical analysis.

| Sub-group analysis | OR (95% CI) | Study heterogeneity ($I^2$; $\tau^2$) | Eligible studies |
|---|---|---|---|
| Additional administration of antibiotics: | | | |
| Allowed | 0.95 (0.59–1.51) | 3%; 0.01 | *Dickstein et al., 2020*; *Durante-Mangoni et al., 2013*; *Gibson et al., 1989*; *Harbarth et al., 2015*; *Iravani et al., 1981*; *Jacobs et al., 1993*; *McCarty et al., 1988*; *Paul et al., 2015*; *Pogue et al., 2021*; *Pujol et al., 2021*; *Rubinstein et al., 1995* |
| Prohibited | 0.19 (0.04–0.98) | 57%; 0.79 | *Gerecht et al., 1989*; *Hultén et al., 1997*; *Wurzer et al., 1997* |
| Pre-resistance against non-administered antibiotics required at study inclusion: | | | |
| Required | 1.07 (0.53–2.18) | 17%; 0.10 | *Dickstein et al., 2020*; *Durante-Mangoni et al., 2013*; *Harbarth et al., 2015*; *Parras et al., 1995*; *Paul et al., 2015*; *Pogue et al., 2021*; *Pujol et al., 2021*; *Walsh et al., 1993* |

*Appendix 6—table 4 Continued on next page*

*Appendix 6—table 4 Continued*

| Sub-group analysis | OR (95% CI) | Study heterogeneity ($I^2$; $\tau^2$) | Eligible studies |
|---|---|---|---|
| | | | *Bender et al., 1979; Black et al., 1982; Chaisson et al., 1997; Cometta et al., 1994; Dawson et al., 2015; Dubé et al., 1997; Fournier et al., 1999; Gerecht et al., 1989; Gibson et al., 1989; Hodson et al., 1987; Hoepelman et al., 1988; Hultén et al., 1997; Iravani et al., 1981; Jacobs et al., 1993; Jo et al., 2021; Macnab et al., 1994; Markowitz et al., 1992; May et al., 1997; McCarty et al., 1988; Miehlke et al., 1998; Parry et al., 1977; Rubinstein et al., 1995; Smith et al., 1999; Stack et al., 1998; Wurzer et al., 1997* |
| No | 0.63 (0.23–1.68) | 78%; 2.57 | |
| **Way of antibiotic administration:** | | | |
| | | | *Bender et al., 1979; Black et al., 1982; Chaisson et al., 1997; Dawson et al., 2015; Dubé et al., 1997; Hultén et al., 1997; Iravani et al., 1981; Jo et al., 2021; Macnab et al., 1994; Miehlke et al., 1998; Stack et al., 1998; Walsh et al., 1993; Wurzer et al., 1997* |
| Orally | 0.37 (0.11–1.23) | 69%; 1.96 | |
| | | | *Dickstein et al., 2020; Durante-Mangoni et al., 2013; Gerecht et al., 1989; Gibson et al., 1989; Hoepelman et al., 1988; Jacobs et al., 1993; Markowitz et al., 1992; Pogue et al., 2021; Pujol et al., 2021; Smith et al., 1999* |
| Intravenously | 1.82 (0.64–5.18) | 66%; 0.90 | |
| Different ways of administration in the treatment arms | 2.12 (0.35–12.79) | 1%; 0.01 | *Hodson et al., 1987; Parry et al., 1977; Paul et al., 2015* |

## ii. Pre-resistance against non-administered antibiotics

Some of the studies we included were focused on the treatment of resistant pathogens. Therefore, we tested whether carriage the of resistance against non-administered antibiotics might affect the development of resistance against administered antibiotics. We identified eight studies requiring pre-resistance, of which five had non-zero events in both treatment arms. For both studies requiring pre-resistance and no pre-resistance, we could not identify a trend favouring more or less antibiotics (*Appendix 6—table 3*, *Appendix 6—table 4*). As multi-drug resistance is an increasing concern it is important to understand if the optimal treatment strategy for pre-resistant pathogens might differ from the one of sensitive pathogens. However, the data of our meta-analysis are not sufficient to answer this question.

## iii. Way of antibiotic administration

The way how antibiotics are administered, e.g., intravenously (IV) or orally, could also impact the development of antibiotic resistance due to different pharmacokinetics and potential differing antibiotic bioavailability (*McCarthy and Avent, 2020*). Therefore, we stratified our studies according to the way antibiotics were administered: orally, or IV in both treatment arms, or the way of administration differed in the treatment arms. We could not identify a harm or benefit in the sub-group analyses of using a higher or a lower number of antibiotics (*Appendix 6—table 3*, *Appendix 6—table 4*).

## Appendix 7

## Meta-regressions and multi-model inference

Additionally, to the subgroup analyses, we also performed meta-regressions for the exploration of the importance of factors potentially affecting our main outcomes. For the meta-regression models, we used the conventional random effects model (model 1 in *Jackson et al., 2018*) due to convergence issues with model 4 and since our sensitivity analysis of the main outcomes showed typically robustness to the model choice (Appendix 4). By performing meta-regressions, we were able to include continuous covariables such as treatment length, and by multi-model inference, we could obtain parameter estimates averaged over a set of models. The set of possible models was restricted to meta-regression models with up to two covariables and no interaction terms to avoid overfitting. We performed multi-model inference with the R package *MuMIn* (version 1.46) (*Bartoń, 2020*).

For the multi-model inference, all meta-regression models of the set of possible models were simulated. Following a model selection approach using the Akaike information criterion (AIC) model, the AIC value for each model was calculated. The AIC is a measure of fit, which is based on the log-likelihood function, and the number of unknown model parameters. Smaller AIC values are assigned to better model fits. In addition to the AIC value, we calculated the AIC differences (ΔAIC) between each model and the model with the lowest AIC value. With ΔAIC we calculated the Akaike weights, which can be interpreted as the probability a model is the best of the given set of models and data. With the full model approach, we then calculated the model-averaged coefficients, which are estimates weighted by the Akaike weights and averaged over the whole set of possible models (full model average). For the interested reader further, detailed information can be found in literature about multi-model inference (*Anderson, 2008*; *Burnham and Anderson, 2002*; *Symonds and Moussalli, 2011*).

The covariables we considered for the meta-regression included: (i) administration of antibiotics common to the treatment arms, (ii) required comorbidity status at study inclusion, (iii) the year difference between the youngest antibiotic in the treatment arm with a lower number of antibiotics and the youngest in the treatment arm with a higher number of antibiotics, (iv) the treatment length, (v) the length of study/resistance follow up, (vi) gram status of bacteria with resistance measurements, and (vii) the number of antibiotics administered.

In some cases, the treatment lengths of the two treatment arms within a study were of different lengths, in those cases, we took as the treatment length covariable the average treatment time of both treatment arms. As the treatment times between treatment arms did not vary a lot, we did not explore those differences further. Furthermore, we wanted to consider the age of antibiotics since the conduction of a trial. There are several ways of how to implement this as a covariable. We decided to take the difference of the youngest antibiotics in both treatment arms, as we expected that novel antibiotics are more likely to be tested in the treatment arm with lower antibiotics.

For our multi-model inference, we excluded the variables considering whether a study was conducted in an ICU, and whether additional drugs were administered as we could not confidently obtain information regarding those variables for more than half of the studies. Due to a high correlation between the administration of antibiotics common to the treatment arms and the same dosage (acquisition of resistance: 0.95, de novo emergence of resistance: 0.91), we excluded the variable same dosage from the meta-regressions.

Our multi-model inference showed that for the acquisition of resistance the most important covariable to include in a meta-regression model to explain some of the observed heterogeneity was whether antibiotics common to the treatments were used or not (*Appendix 7—table 1*). This is in line with our sub-group analysis performed (main text *Figure 3B*). By including the information on whether at least one antibiotic was common to both treatment arms in a meta-regression, we could find a decrease in the estimated heterogeneity ($I^2$=59, no-meta-regression: $I^2$=77), but nevertheless, the heterogeneity remains substantial (*Appendix 7—table 2*). Furthermore, we could confirm once more that a lower number of antibiotics performs better, if in the treatment arms, no common antibiotics are used (*Appendix 7—table 1*, *Appendix 7—table 2*). For de novo emergence of resistance, the multi-model inference did not show any significant covariables (*Appendix 7—table 3*).

Overall, this does not necessarily mean that any of the covariables are not impacting the outcome of resistance development significantly, but since most studies were underpowered (main text *Figure 2B*) there is the possibility that we are missing important signals.

**Appendix 7—table 1.** Overview of the model-averaged coefficients obtained by the multi-model inference for the main outcome acquisition of resistance.

Significant model estimates are displayed in a bold font.

| Model-averaged coefficients (full-average) | Estimated | Standard error | z value | Pr(>\|z\|) |
|---|---|---|---|---|
| Intercept | 0.73 | 1.69 | 0.60 | 0.54 |
| Length of follow-up | 1.00 | 1.00 | 0.46 | 0.65 |
| Treatment length | 0.99 | 0.01 | 0.77 | 0.43 |
| 1 vs 3 antibiotics | 0.78 | 2.19 | 0.31 | 0.75 |
| 2 vs 3 antibiotics | 1.16 | 2.16 | 0.19 | 0.85 |
| Antibiotics in common: no | **5.67** | **1.89** | **2.73** | **0.01** |
| Comorbidity: yes | 1.35 | 1.77 | 0.53 | 0.60 |
| Gram-positive and negative bacteria | 1.52 | 1.91 | 0.65 | 0.52 |
| Gram-positive bacteria | 1.08 | 2.07 | 0.11 | 0.91 |
| Year difference of youngest antibiotics | 1.00 | 1.01 | 0.15 | 0.88 |

**Appendix 7—table 2.** Model output for a meta-regression for acquisition of resistance including as a covariable, whether at least one antibiotic was in common in the treatment arms.

Significant model estimates are displayed in a bold font.

| | OR (95% CI) | z value | Pr(>\|z\|) | Study heterogeneity ($I^2$; $\tau^2$) |
|---|---|---|---|---|
| Intercept | 0.63 (0.33–1.21) | −1.39 | 0.17 | |
| Antibiotics common: no | 5.86 (2.05–16.76) | 3.30 | <0.01 | 59%; 0.90 |

**Appendix 7—table 3.** Overview of the model-averaged coefficients obtained by the multi-model inference for the main outcome de novo emergence of resistance.

| Model-averaged coefficients (full-average) | Estimated | Standard error | z value | Pr(>\|z\|) |
|---|---|---|---|---|
| Intercept | 2.22 | 2.42 | 0.90 | 0.37 |
| Length of follow-up | 0.99 | 1.01 | 0.73 | 0.46 |
| Treatment length | 1.00 | 1.01 | 0.41 | 0.68 |
| 2 vs 3 antibiotics | 1.29 | 2.51 | 0.28 | 0.78 |
| Antibiotics in common: yes | 0.32 | 2.66 | 1.16 | 0.25 |
| Comorbidity: yes | 1.40 | 2.03 | 0.47 | 0.64 |
| Gram-positive and negative bacteria | 1.45 | 2.23 | 0.47 | 0.64 |
| Gram-positive bacteria | 1.16 | 1.98 | 0.21 | 0.83 |
| Year difference of youngest antibiotics | 0.99 | 1.02 | 0.30 | 0.72 |

# Appendix 8

## Statistical power

### 1. Adequate treatment arm size

Resistance development is a rare event and therefore differences in resistance development are difficult to detect in small population sizes. To illustrate this, we calculated how much participants would have needed to be included per treatment harm in order to detect whether a higher number of antibiotics would half the odds of occurrence of resistance and compared it to the actual number of participants (*Appendix 8—figure 1*). For the calculations we assumed a power of 80% and used for each trial the upper confidence interval for the probability of resistance development in the treatment arm with the lower number of antibiotics. The confidence interval was determined with Bayesian inference.

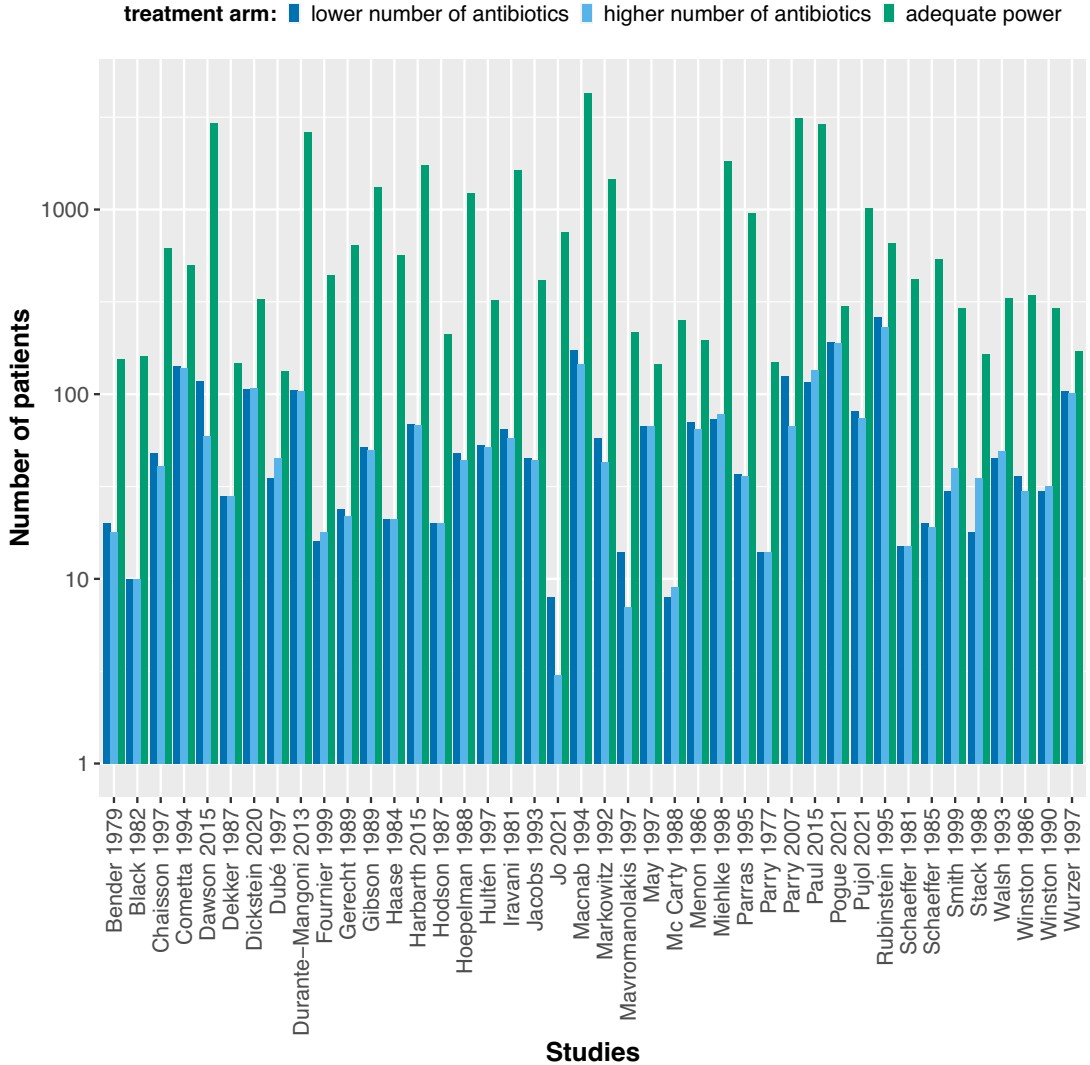

**Appendix 8—figure 1.** The calculated adequate treatment arm size for each study assuming to detect an odds ratio of 0.5 with 80% power in comparison to the actual treatment arm sizes. The power calculations were performed using the upper confidence interval for the binomial probability of the treatment arm with fewer antibiotics.

### 2. Trial sequential analysis

It is expected that pooling data from several RCTs results in a high level of evidence. Nevertheless, meta-analysis might lead to inconclusive results or even misleading ones as a meta-analyses can

also suffer from low statistical power (*Kang, 2021*). Therefore, we performed for our two main outcomes a trial sequential analysis (TSA), using the TSA tool version 0.9.5.10 Beta (Copenhagen: The Copenhagen Trial Unit, Centre for Clinical Intervention Research, 2016) to assess how strong and sufficient the evidence of our overall analyses is. For both outcomes, the TSA supports that the existing evidence on resistance development is not sufficient and conclusive, as the trial sequential monitoring boundary is not crossed by the Z-curve in any of the cases, nor is the required sample size reached (*Appendix 8—figure 2*). For the TSA calculations we used resistance incidence rate per treatment arm, which we calculated by averaging the incidence rates of all included studies (per outcome). For interested readers, technical details of the TSA can be found elsewhere (*Kang, 2021*; *Wetterslev et al., 2017*). The TSA analysis is an additional analysis, which was not predefined in our study protocol.

*Appendix 8—figure 2 continued on next page*

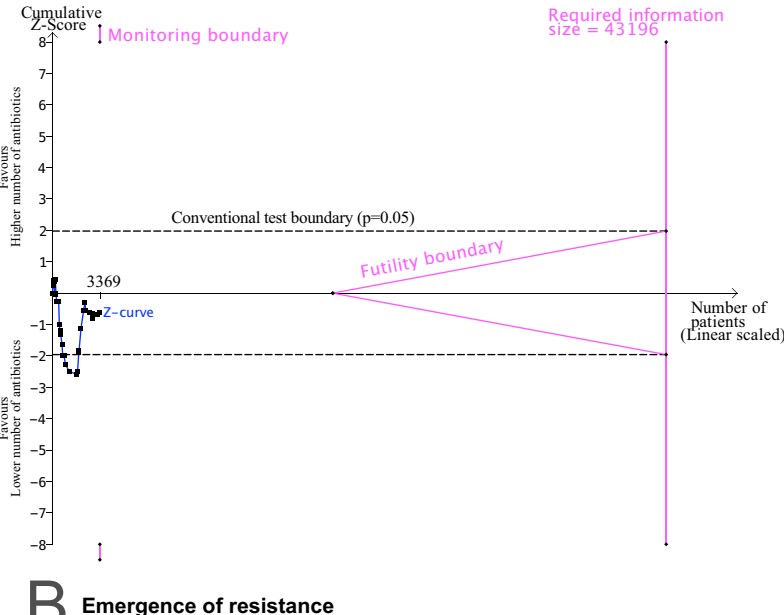

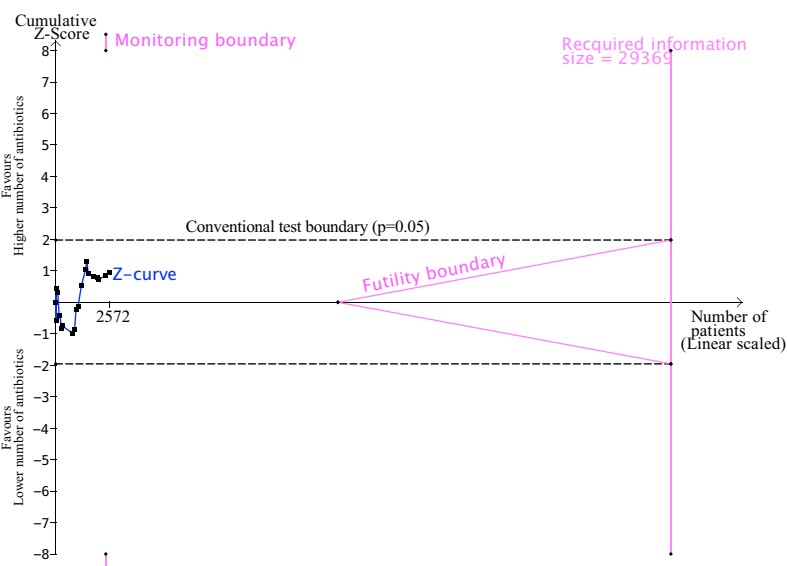

**Appendix 8—figure 2.** Trial sequential analysis (TSA) output using 80% power, and 5% significance to detect a relative odds reduction of 50%: (**A**) acquisition of resistance. (**B**) de novo emergence of resistance. No sufficient evidence on the development of resistance is supported, since the Z-curves do not cross the monitoring nor the futility boundaries, and the required sample size is not reached.

# Appendix 9

## Secondary outcomes

In the evaluation of an optimal antibiotic treatment strategy, many factors play a role besides the potential spread of antibiotic resistance and therewith the future potential to treat infections successfully. One important factor, which is naturally the focus of clinical research, is the wellbeing of the patient receiving antibiotic treatment. Antibiotic combination therapy is often associated with a higher medical burden for the treated patient, e.g., through a higher risk of toxicity (*Tamma et al., 2012*). To present are more comprehensive evaluation of antibiotic combination therapy, we systematically summarised the following outcomes as an indication of the wellbeing of the treated patient: (i) All-cause mortality, (ii) mortality attributable to infection, (iii) treatment failure, (iv) treatment failure due to a change of resistance against the study drugs, and (v) proportion of patients with alterations to the treatment due to adverse events. Additionally, we collected data on acquisition, and de novo emergence of resistance against non-administered antibiotics to further assess the risk of resistance spread, which might affect future treatment success. Overall, we did not find any indication of a difference for most evaluation metrics of combining a higher number of antibiotics in comparison to less as presented below. Only the probability of the secondary outcome 'alterations of the prescribed treatment due to adverse events', was higher using more antibiotics in comparison to fewer.

## i. All-cause mortality

We extracted the number of patients that died in a study as reported. We did not identify a mortality difference in using a higher number of antibiotics as opposed to less (*Appendix 9—figure 1*). One must consider that the estimated pooled OR 0.98 (95% CI 0.79–1.21) was based on several RCTs with different sources of potential heterogeneity, which we did not account for in our statistical analysis of secondary outcomes. Nevertheless, the heterogeneity in our random effects model for all-cause mortality could be classified as unimportant ($I^2$=11%). In previously conducted meta-analyses evaluating antibiotic combination therapy mortality was often the main outcome, but the inclusion criteria were less broad, and constrained to specific diseases, pathogens, or particular antibiotic combinations. The results of those meta-analyses do not easily generalize to one overall trend, but rather highlight that sub-analyses accounting for specific infections and antibiotic comparisons might be important as we found for our main outcomes of resistance development (*Paul et al., 2014*; *Schmid et al., 2019*; *Ye et al., 2021*). Nevertheless, we found in line with most previous meta-analyses no clear harm or benefit of combining a higher number of antibiotics or less with respect to all-cause mortality.

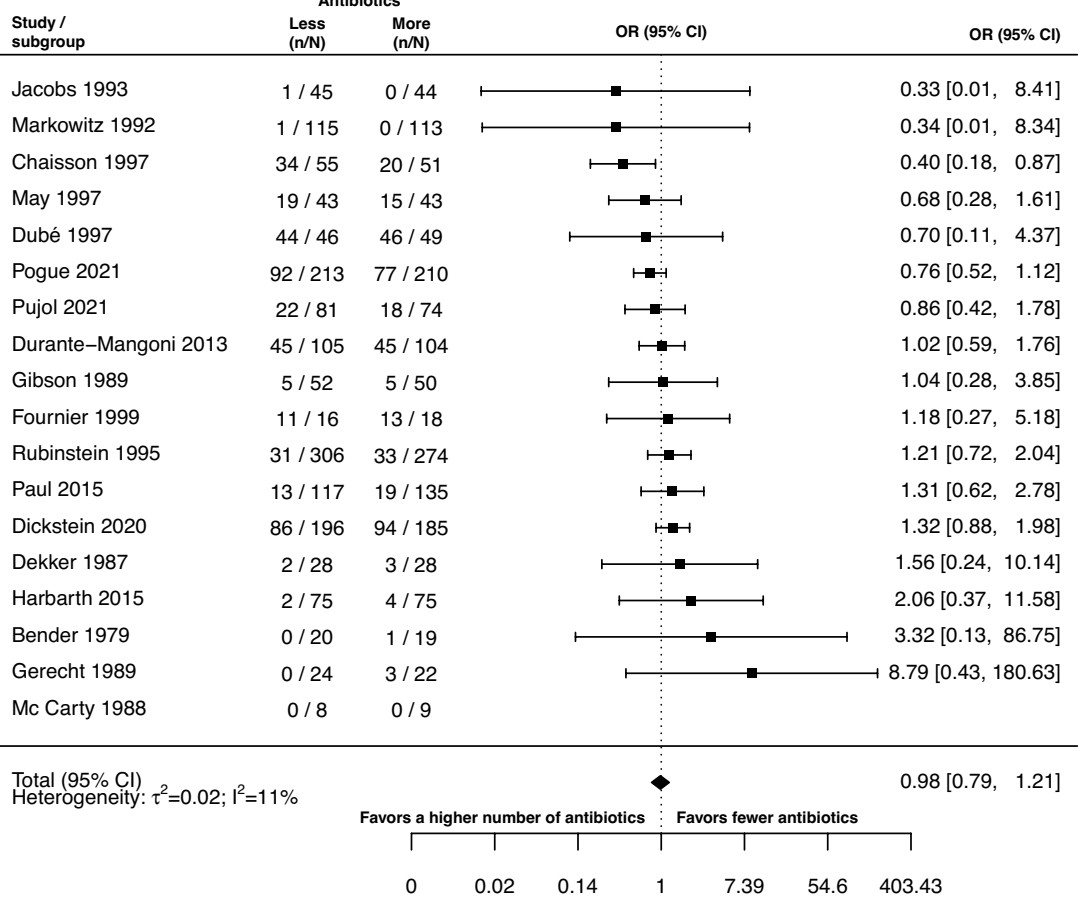

**Appendix 9—figure 1.** Forest plot of all-cause mortality.

## ii. Mortality attributable to infection

Besides all cause-mortality, we also extracted the number of deaths that the respective study authors attributed to the infection treated. As for all-cause mortality our estimate for mortality, attributable to infection indicated no difference between treating with a higher number of antibiotics in comparison to less (pooled OR 1.05, 95% CI 0.64–1.71; *Appendix 9—figure 2*), and the model heterogeneity could also be classified as unimportant ($I^2$=12%).

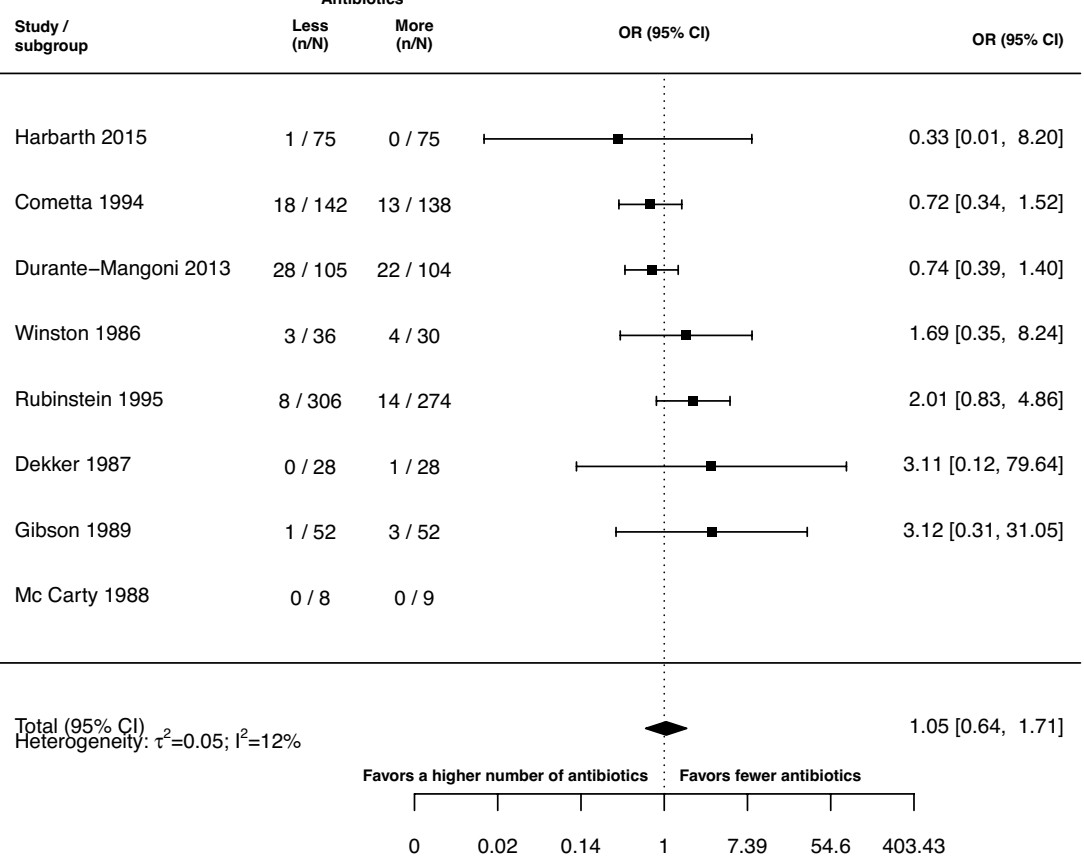

Appendix 9—figure 2. Forest plot of mortality attributable to infection.

### iii. Treatment failure

We extracted the number of treatment failures in each treatment arm if treatment failure was explicitly defined or classified by the study authors. As the selection of studies for this meta-analysis was not restricted to one specific pathogen, or condition requiring antibiotic treatment, we expected a variety of different reasons for the employment of antibiotics. Out of practicality and to account for the different conditions treated, we decided not to pre-define our own criteria for treatment failure for each condition, but rather use the study's author's interpretation of treatment failure (*Appendix 9—table 1*).

Our estimate gave no indication for a difference in treatment failure when treating with a higher number of antibiotics in comparison with a lower number of antibiotics if treatment failure was considered (pooled OR 0.98, 95% CI 0.66–1.47; *Appendix 9—figure 3*). However, we observed a substantial amount of heterogeneity in our model ($I^2$=74%), which might indicate that for some bacterial conditions or some antibiotic combinations, there might be a difference.

Appendix 9—table 1. Oerview of different treatment failure definitions.

| Study | Definition of treatment failure given by the study authors |
|---|---|
| *Cometta et al., 1994* | Lack of improvement of primary infection, development of a sepsis syndrome or septic shock during treatment, superinfection |
| *Durante-Mangoni et al., 2013* | No improvement of clinical conditions by day 21 or worsening of the condition at any time, given persistently positive *Acinetobacter baumannii* cultures |

*Appendix 9—table 1 Continued on next page*

*Appendix 9—table 1 Continued*

| Study | Definition of treatment failure given by the study authors |
|---|---|
| *Gerecht et al., 1989* | Continued presence of infecting organism(s) in bile cultures, with persistent indications of cholangitis, or superinfection, or the presence of new infecting organism(s) during or at the end of antibiotic treatment, with indications of cholangitis, or the emergence of an infecting organism(s) resistant to gentamicin or mezlocillin during treatment, with indications of cholangitis, or the emergence of an infecting organism(s) resistant to gentamicin or mezlocillin during treatment, with indications of cholangitis, or relapse, or recurrence of indications of cholangitis, with the original infecting organism(s) present in cultures of bile or blood within 8 wk after treatment, or death due to uncontrolled infection. |
| *Haase et al., 1984* | The persisting presence of the pretherapy infecting organism, with or without pyuria, during treatment. |
| *Harbarth et al., 2015* | No improvement or worsening in the clinical condition, or a change of the assigned therapy at any time, or death. |
| *Jacobs et al., 1993* | No apparent response to therapy and no definitive identification of an alternative etiology that would explain this lack of response. |
| *Markowitz et al., 1992* | Persistence of septic pulmonary emboli, persistence of positive blood or deep tissue cultures, or relapse after the end of presumably adequate treatment. |
| *May et al., 1997* | Treatment failure was defined as all other situations than success, whereas the primary determinants of success were as follows: patient living, either not fever or a reduction of ≥1 °C in initial body temperature, and a blood culture negative for *M. avium* |
| *Parry et al., 2007* | Continuing fever with at least one other typhoid-related symptom for more than 7 d after the start of treatment, or a required change in therapy due to the development of severe complications during treatment (severe gastrointestinal bleeding, intestinal perforation, visible jaundice, myocarditis, pneumonia, renal failure, shock, or an altered conscious level) |
| *Paul et al., 2015* | Treatment failure at 7 d was defined as a composition of death, persistence of fever, persistence of hypotension, non-improving Sequential Organ Failure Assessment score, or persistent bacteraemia on day 7. |
| *Pogue et al., 2021* | Clinical failure was defined by meeting any of the following criteria: death either during therapy or within 7 d after; receipt of rescue therapy for the trial pathogen within 7 d after treatment, exclusion from the trial due to an adverse event considered related to trial treatment; bacteremia more than 5 d after the begin of therapy for patients with blood stream infections; or failure to improve or worsening of oxygenation by the end of trial treatment in patients with pneumonia. |
| *Pujol et al., 2021* | No clinical improvement after 3 d of therapy, persistent MRSA bacteraemia at day 7 or later, early discontinuation of therapy due to adverse events or based on clinical judgment, recurrent MRSA bacteraemia before or at the test of cure, missing blood cultures at the test of cure, and/or death due to any cause before the test of cure. |
| *Rubinstein et al., 1995* | Use of a new antibiotic due to a worsening in clinical condition, isolation of resistant organism, or superinfection at the initial site during treatment, no clinical response or death attributed to infection. |

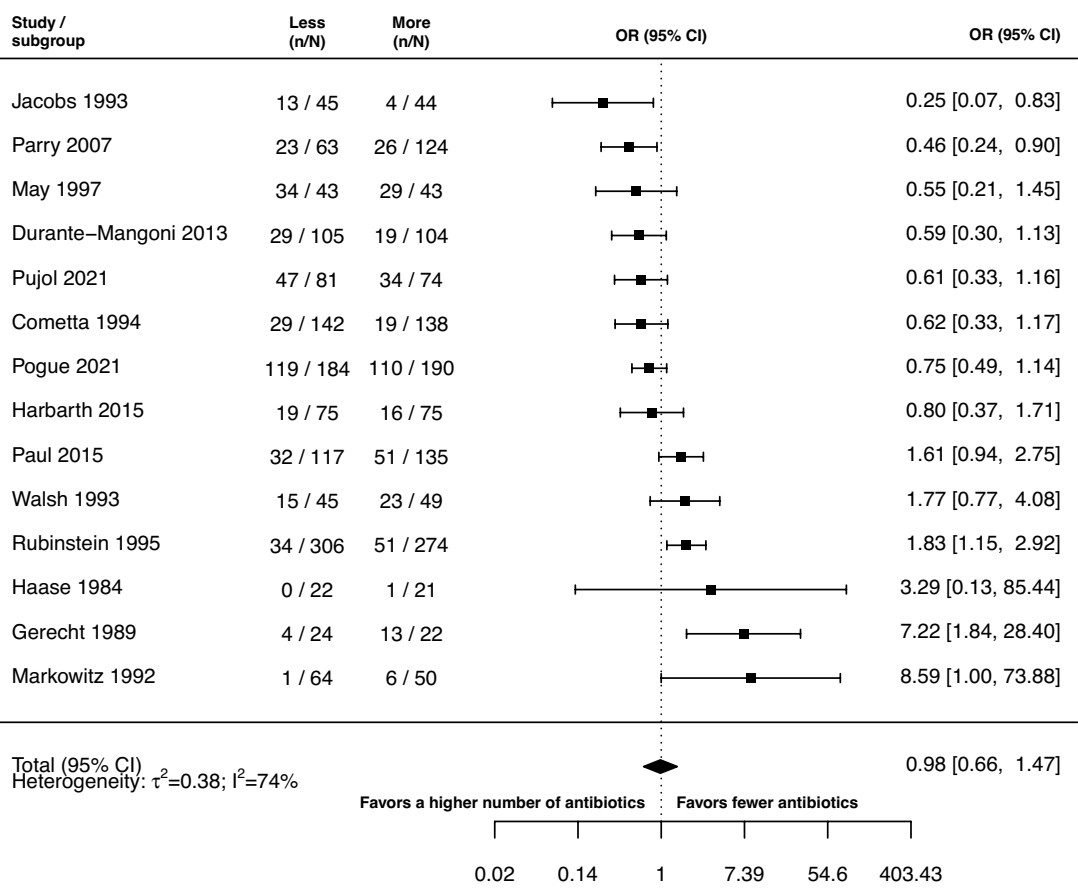

**Appendix 9—figure 3.** Forest plot of treatment failure.

## iv. Treatment failure due to a change of resistance against the study drugs

We could only extract information for treatment failure due to a change of resistance against the study drugs from three out of the 42 studies. As one of the studies had zero-events in both treatment arms our statistical summary estimate was only based on two studies and should, therefore, be interpreted with caution. Nevertheless, as for treatment failure we did not identify a difference of using a higher number of antibiotics in comparison to less when considering treatment failure due to a change of resistance against the study drugs (pooled LOR 0.61, 95% CI 0.29–1.28; $I^2$=1%; *Appendix 9—figure 4*).

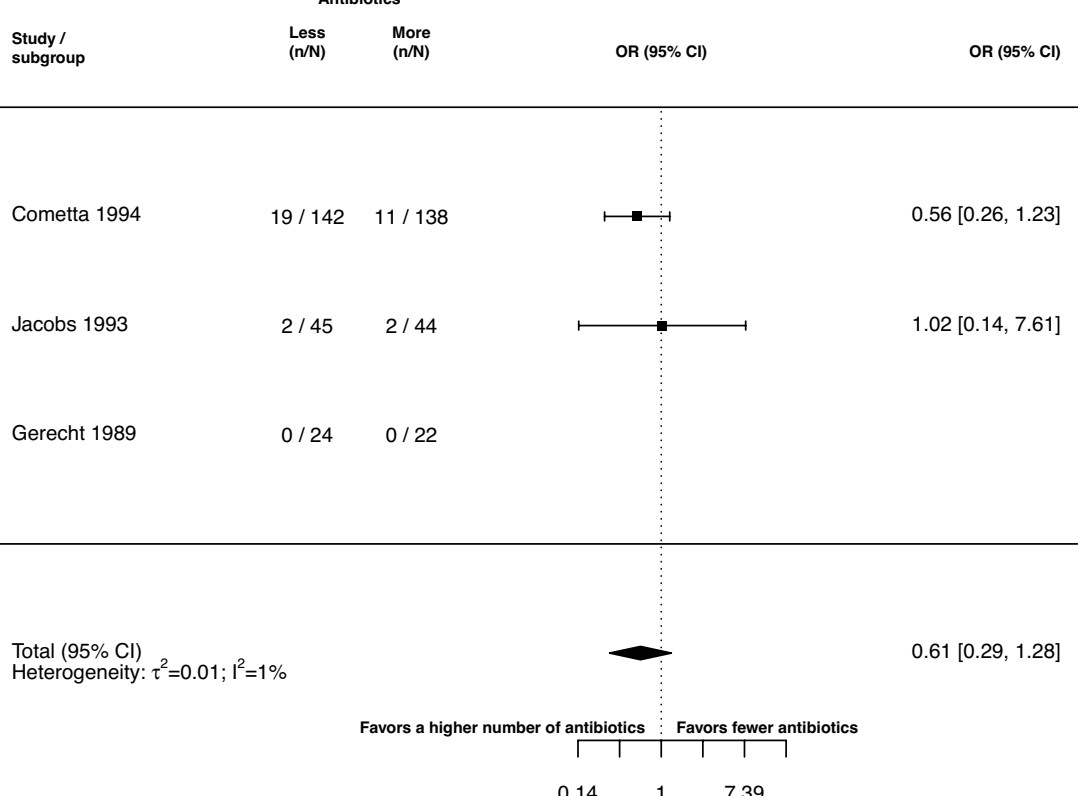

**Appendix 9—figure 4.** Forest plot of treatment failure due to a change of resistance against the study drugs.

## v. Alterations of the prescribed treatment due to adverse events

To get an indication of how well the treatments were tolerated by the patients we extracted data on alterations of the prescribed treatment due to adverse events. We did identify the benefit of using a lower number of antibiotics in comparison to a higher one. The heterogeneity in the random effects model could be classified as unimportant (pooled OR 1.61, 95% CI 1.12–2.31; $I^2$=5%; *Appendix 9— figure 5*).

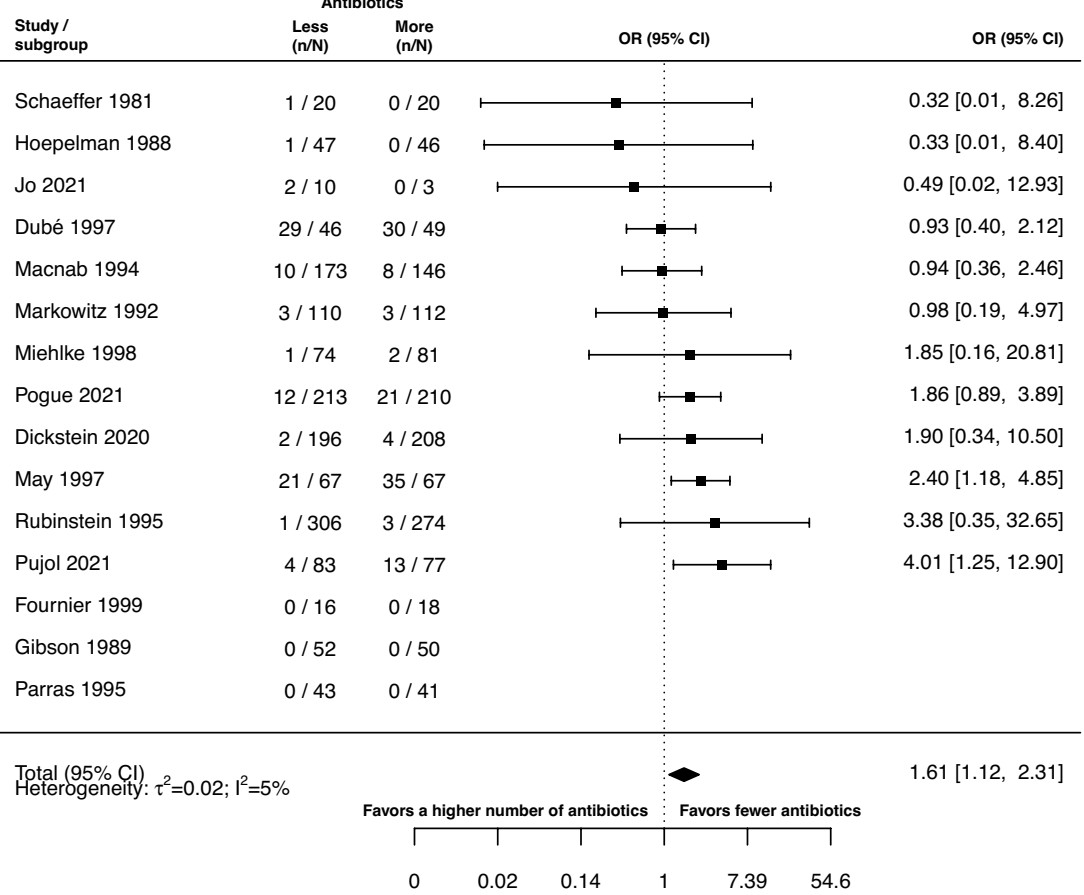

| Study / subgroup | Antibiotics Less (n/N) | More (n/N) | OR (95% CI) | OR (95% CI) |
|---|---|---|---|---|
| Schaeffer 1981 | 1 / 20 | 0 / 20 | | 0.32 [0.01, 8.26] |
| Hoepelman 1988 | 1 / 47 | 0 / 46 | | 0.33 [0.01, 8.40] |
| Jo 2021 | 2 / 10 | 0 / 3 | | 0.49 [0.02, 12.93] |
| Dubé 1997 | 29 / 46 | 30 / 49 | | 0.93 [0.40, 2.12] |
| Macnab 1994 | 10 / 173 | 8 / 146 | | 0.94 [0.36, 2.46] |
| Markowitz 1992 | 3 / 110 | 3 / 112 | | 0.98 [0.19, 4.97] |
| Miehlke 1998 | 1 / 74 | 2 / 81 | | 1.85 [0.16, 20.81] |
| Pogue 2021 | 12 / 213 | 21 / 210 | | 1.86 [0.89, 3.89] |
| Dickstein 2020 | 2 / 196 | 4 / 208 | | 1.90 [0.34, 10.50] |
| May 1997 | 21 / 67 | 35 / 67 | | 2.40 [1.18, 4.85] |
| Rubinstein 1995 | 1 / 306 | 3 / 274 | | 3.38 [0.35, 32.65] |
| Pujol 2021 | 4 / 83 | 13 / 77 | | 4.01 [1.25, 12.90] |
| Fournier 1999 | 0 / 16 | 0 / 18 | | |
| Gibson 1989 | 0 / 52 | 0 / 50 | | |
| Parras 1995 | 0 / 43 | 0 / 41 | | |
| Total (95% CI) Heterogeneity: $\tau^2$=0.02; $I^2$=5% | | | | 1.61 [1.12, 2.31] |

Favors a higher number of antibiotics    Favors fewer antibiotics

0    0.02    0.14    1    7.39    54.6

**Appendix 9—figure 5.** Forest plot of alterations of the prescribed treatment due to adverse events.

## vi. Acquisition of resistance against non-administered antibiotics

There are several ways of how bacteria may get resistant against antibiotics, one of them is through acquiring antibiotic resistance plasmids. Clinically relevant plasmids often confer resistance against multiple antibiotics (*Cazares et al., 2020*; *Holt et al., 2011*; *Paterson and Bonomo, 2005*). Therefore, one might expect if a patient is treated with a higher number of antibiotics the chances increase of acquiring multidrug-resistant plasmids that confer resistance to antibiotics that are not part of the current treatment. In addition, one could expect, that the chances for cross resistances increase, i.e., the obtained resistance confers resistances to several antibiotics, if a higher number of antibiotics is administered. To check this reasoning, we extracted the data for acquisition, and de novo emergence of resistance against non-administered antibiotics.

For seven studies we extracted the data for acquisition of resistance against non-administered antibiotics, but we could only use three of them for our statistical analyses as the other studies had zero events in both treatment arms. As the statistical analysis was only based on three studies and the model showed moderate to substantial heterogeneity ($I^2$=60%) our estimate might not be sufficient to confidently give an indication. The pooled LOR of our random effects model suggested no difference in using a higher number of antibiotics in comparison to less to reduce acquisition of resistance against non-administered drugs (OR 0.39, 95% CI 0.02–8.48; figure *Appendix 9—figure 6*).

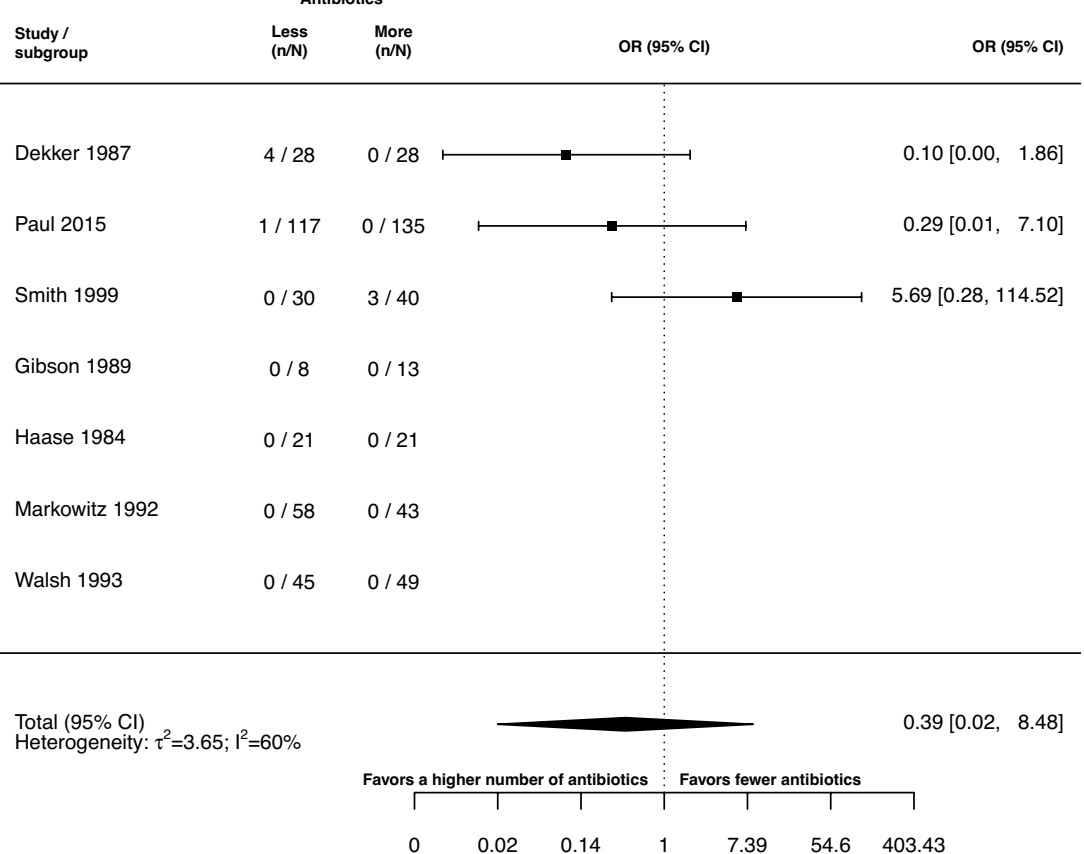

**Appendix 9—figure 6.** Forest plot of acquisition of resistance against non-administered antibiotics.

### vii. De novo emergence of resistance against non-administered antibiotics

As for the main outcomes we distinguished between acquisition and de novo emergence of resistance. According to our definition (main text: Methods, Appendix 1), de novo emergence of resistance is a subset of acquisition of resistance. For the acquisition of resistance against non-administered antibiotics, we obtained three studies eligible for the statistical analysis, and for de novo emergence only two. Therefore, the estimates need to be taken into consideration. As for the acquisition of resistance against non-administered the antibiotics there was no indication of a difference in using a higher or a lower number of antibiotics (pooled OR 1.91, 95% CI 0.09–39.69; $I^2$=33%; *Appendix 9—figure 7*)

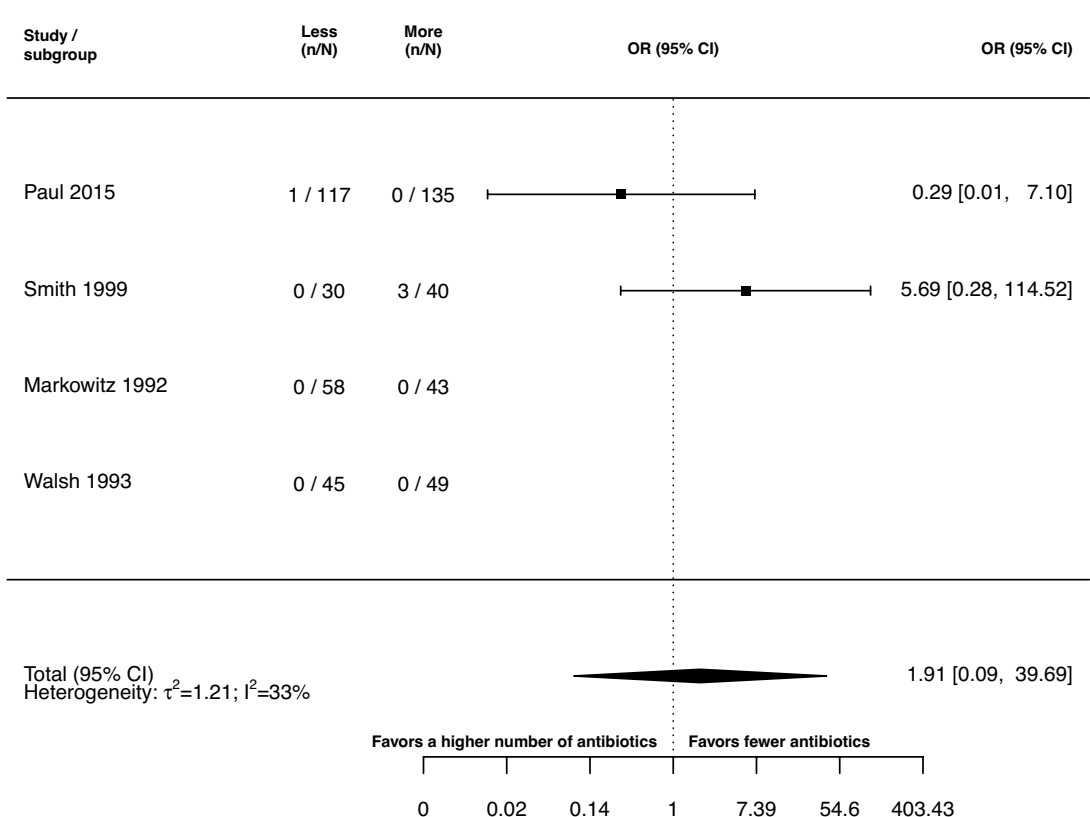

**Appendix 9—figure 7.** Forest plot of de novo emergence of resistance against non-administered antibiotics.

## Appendix 10

### List of contacted authors and reasoning for exclusion of studies included in previous meta-analyses

An overview of authors, that were contacted for clarification of study data, is shown in *Appendix 10—table 1*. In our meta-analysis we excluded some studies that were included in previous meta-analyses focusing on resistance development (*Bliziotis et al., 2005*; *Paul et al., 2014*). An overview of those studies and an exclusion reason is given in table *Appendix 10—table 2*.

**Appendix 10—table 1.** List of studies for which study authors or institutions were contacted.
An indication is given as to whether clarifying information was obtained.

| Study | Person/Institution contacted | Information sufficient for paper inclusion obtained (yes/no) |
| --- | --- | --- |
| *African and Councils, 1972* | Research office of the Royal Brompton & Harefield hospitals | no |
| *Bazzoli et al., 1998* | Franco Bazolli | no |
| *Benson et al., 2000* | Constance Benson | no |
| *Bochenek et al., 2003* | David Yates Graham; Wieslaw Bochenek | no |
| *Bosso and Black, 1988* | John Bosso | no |
| *Bow et al., 1987* | Eric Bow | no |
| *Cruciani et al., 1989* | Mario Cruciani | no |
| *Dalgic et al., 2014* | Nazan Dalgic | no |
| *de Pauw et al., 1985* | Ben de Pauw | no |
| *de Pauw, 1987* | Ben de Pauw | no |
| *DiNubile et al., 2005* | Mark Dinubile | no |
| *Frank et al., 2002* | Elliot Frank | no |
| *Gold et al., 1985* | Ronald Gold | no |
| *Grossman et al., 1994* | Ronald Grossman | no |
| *Grabe et al., 1986* | Magnus Grabe | no |
| *Guerrant et al., 1981* | Richard Guerrant | no |
| *Heyland et al., 2008* | Daren Heyland | no |
| *Hodson et al., 1987* | Margaret Hodson | no |
| *Hoepelman et al., 1988* | Andy I.M. Hoepelman | no |
| *Jackson et al., 1986* | Mary Anne Jackson | no |
| *Liang et al., 1990* | Raymond Hin Suen Liang | no |
| *McLaughlin et al., 1983* | John McLaughlin | no |
| *Muder et al., 1994* | Robert Muder | no |
| *Padoan et al., 1987* | Rita Padoan | no |
| *Paul et al., 2015* | Mical Paul | yes |
| *Parry et al., 2007* | Christopher Parry | yes |
| *Pujol et al., 2021* | Miquel Pujol | yes |
| *Schaad et al., 1997* | Urs Schaad | no |
| *Shawky et al., 2022* | Sherief Bad-Elsalam | no |
| *Sun et al., 2022* | Jia Fan | no |

**Appendix 10—table 2.** Table of studies, which were included in previous meta-analyses, but excluded in our study.
The reason for exclusion is indicated. *In our protocol we stated, that we would include articles in the Russian language. However, since VNK, the only Russian-speaking author, did not screen all the papers from our systematic search for inclusion, we excluded studies in the Russian language.

| Study | Inclusion in previous meta-analyses | Reason for exclusion | Identified with our search strategy |
|---|---|---|---|
| *Carbon et al., 1987* | *Paul et al., 2014* | Not accessible via ETH Zurich library services | no |
| *Cone et al., 1985* | *Blaziotis et al., 2005* | No data on resistance emergence, due to no clear statement of how many resistances are measured in the treatment arm with more antibiotics | no |
| *Croce et al., 1993* | *Blaziotis et al., 2005* | No proper randomisation of treatment strategies, i.e., the trial was conducted in different phases | yes |
| *German and Austrian Imipenem/ Cilastatin Study Group, 1992* | *Blaziotis et al., 2005, Paul et al., 2014* | No fixed treatment, as an additional antibiotic was allowed to be administered only in the treatment arm with more antibiotics | no |
| *Gribble et al., 1983* | *Blaziotis et al., 2005* | No fixed treatment, since antibiotics could be substituted during treatment | yes |
| *Iakovlev et al., 1998* | *Paul et al., 2014* | Russian language* | no |
| *Klastersky et al., 1973* | *Paul et al., 2014* | Not clearly extractable how many patients developed resistance | no |
| *Mandell et al., 1987* | *Blaziotis et al., 2005, Paul et al., 2014* | Treatment is not fixed due to alterations of treatment based on the infecting organism | no |
| *Sculier et al., 1982* | *Paul et al., 2014* | No proper comparison, since the study does not compare per se a different number of antibiotics but adds an additional way of administration of the same antibiotic | no |

## Appendix 11

### 1. Search strategy

#### a) PubMed

(((((((((((((('Bacterial Infections/Drug Therapy'[mesh]) OR 'Bacterial Infections/drug effects'[Mesh]) OR 'Bacteria/drug effects'[Mesh]) OR 'Bacteria/Drug Therapy'[mesh]) OR (((infection[tiab] OR infections[tiab]) AND bacteria*)))) AND (((((((((((((((('beta-Lactams/Administration and Dosage'[mesh] OR 'beta- Lactams/Therapeutic Use'[mesh])) OR ('Aminoglycosides/Administration and Dosage'[mesh] OR 'Aminoglycosides/Therapeutic Use'[mesh])) OR ('Chloramphenicol/ Administration and Dosage'[mesh] OR 'Chloramphenicol/Therapeutic Use'[mesh])) OR ('Glycopeptides/Administration and Dosage'[mesh] OR 'Glycopeptides/Therapeutic Use'[mesh])) OR ('Rifamycins/Administration and Dosage'[mesh] OR 'Rifamycins/Therapeutic Use'[mesh])) OR ('Streptogramins/Administration and Dosage'[mesh] OR 'Streptogramins/Therapeutic Use'[mesh])) OR ('Sulfonamides/Administration and Dosage'[mesh] OR 'Sulfonamides/Therapeutic Use'[mesh])) OR ('Tetracyclines/Administration and Dosage'[mesh] OR 'Tetracyclines/Therapeutic Use'[mesh])) OR ('Macrolides/Administration and Dosage'[mesh] OR 'Macrolides/Therapeutic Use'[mesh])) OR ('Oxazolidinones/Administration and Dosage'[mesh] OR 'Oxazolidinones/Therapeutic Use'[mesh])) OR ('QUINOLONES/Administration and Dosage'[mesh] OR 'QUINOLONES/Therapeutic Use'[mesh])) OR ('Lipopeptides/Administration and Dosage'[mesh] OR 'Lipopeptides/Therapeutic Use'[mesh])) OR ('Anti-Bacterial Agents/Administration and Dosage'[mesh:noexp]))) OR 'Anti-Bacterial Agents/Therapeutic Use'[mesh:noexp]) OR 'Anti-Bacterial Agents/Therapy'[mesh:noexp]) OR antibiotic*[tiab])) AND (((((((('Drug Therapy, Combination'[mesh:noexp]) OR 'drug combinations'[mesh:noexp]) OR 'trimethoprim, sulfamethoxazole drug combination'[mesh:noexp]) OR 'Drug Synergism'[mesh:noexp]))) OR (combination[tiab] AND (therapy[tiab] OR therapies[tiab]))) OR combinationtherap*[tiab])) AND ((('Drug Resistance, Bacterial'[Mesh]) OR 'Drug Resistance, Microbial'[Mesh:noexp]) OR resistan*[tiab]))) NOT (((('Complementary Therapies'[Mesh]) OR 'Plant Extracts'[Mesh]) OR bismuth[tiab]) OR 'Bismuth'[Mesh]))) AND 'Controlled Clinical Trial'[Publication Type]

#### b) CENTRAL

#1 MeSH descriptor: [Bacterial Infections] explode all trees and with a qualifier(s): [drug therapy - DT]

#2 MeSH descriptor: [Bacteria] explode all trees and with qualifier(s): [drug effects -

#3 ((infection):ti,ab,kw OR (infections):ti,ab,kw) AND bacteria*

#4 MeSH descriptor: [beta-Lactams] explode all trees and with a qualifier(s): [administration & dosage - AD, therapeutic use - TU]

#5 MeSH descriptor: [Chloramphenicol] explode all trees and with qualifier(s): [administration & dosage - AD, therapeutic use - TU]

#6 MeSH descriptor: [Aminoglycosides] explode all trees and with qualifier(s): [administration & dosage - AD, therapeutic use - TU]

#7 MeSH descriptor: [Glycopeptides] explode all trees and with qualifier(s): [administration & dosage - AD, therapeutic use - TU]

#8 MeSH descriptor: [Rifamycins] explode all trees and with qualifier(s): [administration & dosage - AD, therapeutic use - TU]

#9 MeSH descriptor: [Streptogramins] explode all trees and with a qualifier(s): [administration & dosage - AD, therapeutic use - TU]

#10 MeSH descriptor: [Sulfonamides] explode all trees and with qualifier(s): [administration & dosage - AD, therapeutic use - TU]

#11 MeSH descriptor: [Macrolides] explode all trees and with qualifier(s): [administration & dosage - AD, therapeutic use - TU]

#12 MeSH descriptor: [Tetracyclines] explode all trees and with qualifier(s): [administration & dosage - AD, therapeutic use - TU]

#13 MeSH descriptor: [Oxazolidinones] explode all trees and with qualifier(s): [administration & dosage - AD, therapeutic use - TU]

#14 MeSH descriptor: [Quinolones] explode all trees and with qualifier(s): [administration & dosage - AD, therapeutic use - TU]

#15 MeSH descriptor: [Lipopeptides] explode all trees and with qualifier(s): [administration & dosage - AD, therapeutic use - TU]

#16 MeSH descriptor: [Anti-Bacterial Agents] this term only and with qualifier(s): [administration & dosage - AD, therapeutic use - TU]

#17 (antibiotic*):ti,ab,kw

#18 MeSH descriptor: [Drug Therapy, Combination] This term only

#19 MeSH descriptor: [Drug Combinations] This term only

#20 MeSH descriptor: [Trimethoprim, Sulfamethoxazole Drug Combination] This term only

#21 MeSH descriptor: [Drug Synergism] This term only

#22 ((combination):ti,kw,ab) NEAR/3 ((therapy):ti,kw,ab OR (therapies):ti,ab,kw)

#23 (combinationtherap*):ti,ab,kw

#24 MeSH descriptor: [Drug Resistance, Bacterial] explode all trees

#25 MeSH descriptor: [Drug Resistance, Microbial] this term only

#26 (resistan*):ti,ab,kw

#27 MeSH descriptor: [Complementary Therapies] explode all trees

#28 MeSH descriptor: [Plant Extracts] explode all trees

#29 (bismuth):ti,ab,kw

#30 MeSH descriptor: [Bismuth] explode all trees

#31 {OR #1-#3}

#32 {OR #4-#17}

#33 {OR #18-#23}

#34 {OR #24-#26}

#35 {AND #31-#34}

#36 {OR #27-#30}

#37 #35 NOT #36

## c) EMBASE

#26. #24 AND #25

#25. 'controlled clinical trial'/exp

#24. #23 NOT #22

#23. #18 AND #19 AND #20 AND #21

#22. #14 OR #15 OR #16 OR #17

#21. #12 OR #13

#20. #7 OR #8 OR #9 OR #10 OR #11

#19. #5 OR #6

#18. #1 OR #2 OR #3 OR #4

#17. 'herbal medicine'/exp

#16. 'alternative medicine'/exp

#15. 'bismuth'/exp

#14. bismuth:ti,ab,kw

#13. resistan*:ti,ab,kw

#12. 'antibiotic sensitivity'/exp

#11. (combination NEAR/3 (therapy OR therapies)):ti,ab,kw

#10. combinationtherap*:ti,ab,kw

#9. 'antibiotic agent'/exp/dd_cb

#8. 'drug potentiation'/de

#7. 'combination drug therapy'/de

#6. 'antibiotic*':ti,ab,kw

#5. 'antibiotic agent'/exp

#4. (infection:ti,ab,kw OR infections:ti,ab,kw) AND bacteria*

#3. 'bacterial infection'/exp

#2. 'bacterium'/exp

#1. 'prokaryotes by outer appearance'/exp

## 2. Screening of eligible trials and previous meta-analyses

In addition to the systematic database search, we also screened the references of eligible studies and the trials included in two previous meta-analyses (*Bliziotis et al., 2005*; *Paul et al., 2014*). With the database search, we identified 41 studies. While screening the references of those 41 studies we identified one additional study (*Winston et al., 1986*), which meets our inclusion criteria. This additional study was not identified in our search strategy as neither the abstract nor database specific identifiers gave any indication that resistance was measured in this study. The screening of the trials included in two previous analyses did not result in an inclusion of further studies (*Appendix 10—table 2*).

