## [Editor Report · eLife assessment]

This is a methodologically state-of-the-art systematic review and meta-analysis of studies that addressed the question of whether the administration of multiple antibiotics simultaneously prevents antibiotic resistance development in individuals. The findings are **solid**. Rather than providing a precise answer, the synthesis of studies eligible for analysis leads to the conclusion that "our analysis could not identify any benefit or harm of using a higher or a lower number of antibiotics regarding within-patient resistance development." This article is **important** as it articulates the existing knowledge gap, but also serves as an example for careful future use of the meta-analysis methodology, when existing data just don't allow conclusions.

---

## [Referee Report · Reviewer #2 (Public Review)]

Summary:

The authors performed a systematic review and meta-analysis to investigate whether the frequency of emergence of resistance is different if combination antibiotic therapy is used compared to fewer antibiotics. The review shows that there is currently insufficient evidence to reach a conclusion due to the limited sample size. High-quality studies evaluating appropriate antimicrobial resistance endpoints are needed.

Strengths:

The strength of the manuscript is that the article addresses a relevant research question which is often debated. The article is well-written and the methodology used is valid. The review shows that there is currently insufficient evidence to reach a conclusion due to the limited sample size. High-quality studies evaluating appropriate antimicrobial resistance endpoints are needed. I have several comments and suggestions for the manuscript.

Weaknesses:

Weaknesses of the manuscript are the large clinical and statistical heterogeneity and the lack of clear definitions of acquisition of resistance. Both these weaknesses complicate the interpretation of the study results.

Comments on latest version:

The authors adressed all the comments that were shared in the previous peer review. I still believe that both clinical and statistical heterogeneity remains a problem with the interpretation of the meta-analysis. However, as the authors state, this is in line with the original research question as formulated on Prospero.

---

## [Author Response]

The following is the authors’ response to the original reviews.

**Reviewer #1 (Public Review):**
Major comments:My main concern about the manuscript is the extent of both clinical and statistical heterogeneity, which complicates the interpretation of the results. I don't understand some of the antibiotic comparisons that are included in the systematic review. For instance the study by Paul et al (50), where vancomycin (as monotherapy) is compared to co-trimoxazole (as combination therapy). Emergence (or selection) of co-trimoxazole in *S. aureus* is in itself much more common than vancomycin resistance. It is logical and expected to have more resistance in the co-trimoxazole group compared to the vancomycin group, however, this difference is due to the drug itself and not due to co-trimoxazole being a combination therapy. It is therefore unfair to attribute the difference in resistance to combination therapy. Another example is the study by Walsh (71) where rifampin + novobiocin is compared to rifampin + co-trimoxazole. There is more emergence of resistance in the rifampin + co-trimoxazole group but this could be attributed to novobiocin being a different type of antibiotic than co-trimoxazole instead of the difference being attributed to combination therapy. To improve interpretation and reduce heterogeneity my suggestion would be to limit the primary analyses to regimens where the antibiotics compared are the same but in one group one or more antibiotic(s) are added (i.e. A versus A+B). The other analyses are problematic in their interpretation and should be clearly labeled as secondary and their interpretation discussed.

Thank you for raising these important points and highlighting the need for clarification. We understand that the reviewer has concerns regarding the following points:

(1) The structure of presenting our analyses, i.e. main analyses and sub-group analyses and their corresponding discussion and interpretation

Our primary interest was whether combining antibiotics has an overarching effect on resistance and to identify factors that explain potential differences of the effect of combining antibiotic across pathogens/drugs. Therefore, pooling all studies, and thereby all combinations of antibiotics, is one of our main analyses. The decision to pool all studies that compare a lower number of antibiotics to a higher number of antibiotics was hence predefined in our previously published study protocol (PROSPERO CRD42020187257).

We indeed, find that heterogeneity is high in our statistical analyses. As planned in our study protocol, we did perform several prespecified sub-group analyses and added additional ones. We now emphasize that several sub-group analyses were performed to investigate heterogeneity (L 119ff): “The overall pooled estimates are based on studies that focus on various clinical conditions/pathogens and compare different antibiotics treatments. To explore the impact of these and other potential sources of heterogeneity on the resistance estimates we performed various sub-group analyses and metaregression.”

The performed sub-group analyses specifically focused on specific pathogens/clinical conditions (figure 3) or explored heterogeneity due to different antibiotics in comparator arms – as suggested by the reviewer (figure 3B, SI section 6). We find that the heterogeneity remains high even if only resistances to antibiotics common to both arms are considered (SI section 6.1.8). With this analysis we excluded comparisons of different antibiotics (e.g., A vs B+C), such as those between vancomycin and cotrimoxazole named by the reviewer. While we aimed to explore heterogeneity and investigate potential factors affecting the effect of combining antibiotic on resistance, limitations arose due to limited evidence and the nature of data provided by the identified studies. Therefore, interpretability remains also limited for the subgroup analyses, which we highlight in the discussion. (L 186 ff: We accounted for many sources of heterogeneity using stratification and meta-regression, but analyses were limited by missing information and sparse data.) Further, specific subgroup analyses are discussed in more detail in the SI.

(2) Difference in resistance development due to the type of the antibiotics or due to combination therapy?

The reviewer raises an important point, which we also try to make: future studies should be systematically designed to compare antibiotic combination therapy, i.e. identical antibiotics in treatment arms should be used, except for additional antibiotics used in both treatment arms. We already mentioned this point in our discussion but highlight this now by emphasizing how many studies did not have identical antibiotics in their treatment arms. We write in L194ff: “19 (45%) of our included studies compared treatment arms with no antibiotics in common, and 22 studies (52%) had more than one antibiotic not identical in the treatment arms (table 1). To better evaluate the effect of combination therapy, especially more RCTs would be needed where the basic antibiotic treatment is consistent across both treatment arms, i.e. the antibiotics used in both treatment arms should be identical, except for the additional antibiotic added in the comparator arm (table 1).”

Furthermore, we investigated the importance of the type of antibiotics with several subgroup analyses (e.g. SI sections 6.1.8 and 6.1.10). We now further highlight the concern of the type of antibiotics in the result section of the main manuscript, where we discuss the sub-group analysis with no common antibiotics in the treatment arms 131 ff: “Furthermore, a lower number of antibiotics performed better than a higher number if the compared treatment arms had no antibiotics in common (pooled OR 4.73, 95% CI 2.14 – 10.42; *I2*=37%, SI table S3), which could be due to different potencies or resistance prevalences of antibiotics as discussed in SI (SI section 6.1.10).” As mentioned above we also perform sub-group analyses, where only resistances of antibiotics common to both arms are considered (SI section 6.1.8). However, as discussed in the corresponding sections, the systematic assessment of antibiotic combination therapy remains challenging as not all resistances against antibiotics used in the arms were systematically measured and reported. Furthermore, the power of these sub-group analyses is naturally a concern, as they include fewer studies.

Another concern is about the definition of acquisition of resistance, which is unclear to me. If for example meropenem is administered and the follow-up cultures show Enterococcus species (which is intrinsically resistant to meropenem), does this constitute acquisition of resistance? If so, it would be misleading to determine this as an acquisition of resistance, as many people are colonized with Enterococci and selection of Enterococci under therapy is very common. If this is not considered as the acquisition of resistance please include how the acquisition of resistance is defined per included study. Table S1 is not sufficiently clear because it often only contains how susceptibility testing was done but not which antibiotics were tested and how a strain was classified as resistant or susceptible.

Thank you for pointing out this potential ambiguity. The definition of acquisition of resistance reads now (L 275 ff): “A patient was considered to have acquired resistance if, at the follow-up culture, a resistant bacterium (as defined by the study authors) was detected that was not present in the baseline culture.” We also changed the definition accordingly in the abstract (L 36 ff). We hope that the definition of acquisition is now clearer. Our definition of “acquisition of resistance” is agnostic to bacterial species and hence intrinsically resistant species, as the example raised by the reviewer, can be included if they were only detected during the follow-up culture by the studies. Generally, it was not always clear from the studies, which pathogens were screened for and whether the selection of intrinsically resistant bacteria was reported or not. Therefore, we rely on the studies' specifications of resistant and non-resistant without further distinction from our side, i.e. classifying data into intrinsic and non-intrinsic resistance. Overall, the outcome “acquisition of resistance” can be interpreted as a risk assessment for having any resistant bacterium during or after treatment. In contrast, the outcome “emergence of resistance” is more rigorous, demanding the same species to be detected as more resistant during or after treatment.

The information, which antibiotic susceptibility tests were performed in each individual study can be found in the main text in table 1. However, we agree that this information should be better linked and highlighted again in table S1. We therefore now refer to table 1 in the table description of table S1. L134 ff.: “See table 1 in the main text for which antibiotics the antibiotics tested and reported extractable resistance data”. Furthermore, we added the breakpoints for resistant and susceptible classification if specifically stated in the main text of the study. However, we did not do further research into old guidelines, manufactures manuals or study protocols in case the breakpoints are not specifically stated in the main text as the main goal of this table, in our opinion, is to show a justification, why the studies could be considered for a resistance outcome. We therefore decided against further breakpoint investigations for studies, where the breakpoint is not specifically stated in the main text.

Line 85: "Even though within-patient antibiotic resistance development is rare, it may contribute to the emergence and spread of resistance."Depending on the bug-drug combination, there is great variation in the propensity to develop within-patient antibiotic resistance. For example: within-patient development of ciprofloxacin resistance in Pseudomonas is fairly common while within-patient development of methicillin resistance in *S. aureus* is rare. Based on these differences, large clinical heterogeneity is expected and it is questionable where these studies should be pooled.

We agree that our formulation neglects differences in prevalence of within-host resistance emergence depending on bug-drug combinations. We changed our statement in L 86 to: “Within-patient antibiotic resistance development, even if rare, may contribute to the emergence and spread of resistance.”

Line 114: "The overall pooled OR for acquisition of resistance comparing a lower number of antibiotics versus a higher one was 1.23 (95% CI 0.68 - 2.25), with substantial heterogeneity between studies (I2=77.4%)"What consequential measures did the authors take after determining this high heterogeneity? Did they explore the source of this large heterogeneity? Considering this large heterogeneity, do the authors consider it appropriate to pool these studies?

Thank you for highlighting this lack of clarity. As mentioned above, we now highlight that we performed several subgroup analyses to investigate heterogeneity. (L 116ff): “The overall pooled estimates are based on studies that focus on various clinical conditions/pathogens and compare different antibiotics treatments. To explore the impact of these and other potential sources of heterogeneity on the resistance estimates we performed various subgroup analyses and meta-regression.” Nevertheless, these analyses faced limitations due to the scarcity of evidence and often still showed a high amount of heterogeneity. Given the lack of appropriate evidence, it is hard to identify the source of heterogeneity. The decision to pool all studies was pre-specified in our previously published study protocol (PROSPERO CRD42020187257) and was motivated by the question whether there is a general effect of combination therapy on resistance development or identify factors that explain potential differences of the effect of combination therapy across bug-drug combinations. Therefore, we think that the presentation of the overall pooled estimate is appropriate, as it was predefined, and potential heterogeneity is furthermore explored in the subgroup analyses.

**Reviewer #1 (Recommendations For The Authors):**
I want to congratulate the investigators for the rigorous approach followed and the - in my opinion - correct interpretation of the data and analysis. The disappointing outcome is independent of the quality of the approach used. Yet, the consequences of that outcome are rather limited, and will not be surprising for - at least - some in the field of antibiotic resistance.

Thank you for your positive and differentiated feedback.

**Reviewer #2 (Recommendations For The Authors):**
Line 93: "The screening of the citations of the 41 studies identified one additional eligible study, for a total of 42 studies".Why was this study missed in the search strategy?What is the definition of "quasi-RCTs"? Why were these included in the analysis?

Thank you for pointing out this lack of clarity. The additional study, which was found through screening the references of included studies, was not identified with our search strategy as neither the abstract nor database specific identifiers provided any indications that resistance was measured in this study. We added an explanation in the supplementary materials L 792 ff. and refer to this explanation in the main manuscript (L 95).

Quasi-randomized trials are trials that use allocation methods, which are not considered truly random. We added this specification in L 95. It now reads: “….two quasi-RCTs, where the allocation method used is not truly random” and in L 252 ff: “Studies were classified as quasi-RCTs if the allocation of participants to study arms was not truly random.” For instance, the study Macnab et al. (1994) assigned patients alternately to the treatment arms. Quasi-randomized controlled trials can lead to biases and especially old studies are more likely to have used quasi-random allocation methods. This can also be seen in our study, where the two quasi-randomized controlled trials were published in 1994 and 1997. The bias is considered in the risk of bias assessment and in our conducted sensitivity analysis regarding the impact of risk of bias on our estimates (supplementary information sections 3.0 and 4.2). Furthermore, one of the two previous conducted meta-analyses comparing beta-lactam monotherapy to beta-lactam and aminoglycoside, which assessed resistance development also included quasi-randomized controlled trials Paul et al 2014. Overall, while designing the study, we decided to include quasi-randomized controlled trials to increase statistical power as we expected that limited statistical power might be a concern and decided to assess potential biases in the risk of bias assessment.

Line 100: "Consequently, most studies did not have the statistical power to detect a large effect on within-patient resistance development (figure 2 B, SI p 14).".Small studies actually have more power to detect large effects while smaller power to detect small effects. Please rephrase.

Thank you for pointing out this lack of clarity. We rephrased the sentence in order to emphasize our point that the studies are underpowered even if we assume in our power analysis a large effect on resistance development between treatment arms. In this context “the small” studies include too few patients to detect a large difference in resistance development. As resistance development is a rare event, generally studies have to include a larger number of patients to estimate the effect of intervention. We rephrased the sentence in L 101ff to: “Consequently, most studies did not have the statistical power to detect differences in within-patient resistance development even if we assume that the effect on resistance development is large between treatment arms.”

Line 108: "... and prophylaxis for blood cancer patients with four studies (10%), respectively.".I would suggest using the medical term hematological malignancy patients.

Thank you for the suggestion, we changed it as suggested to hematological malignancy patients, also accordingly in the figures, and table 1.

Line 117: "Since the results for the two resistance outcomes are comparable, our focus in the following is on the acquisition of resistance".The first OR is 1.23 and the second is 0.74, why do you consider these outcomes as comparable?

Thank you for pointing out our unprecise formulation. Due to the lack of power the exact estimates need to be interpreted with care. Here, we wanted to make the point that qualitatively the results of both outcomes do not differ in the sense that our analysis shows no substantial difference between a higher and a lower number of antibiotics. We rephrased the sentence to be more precise (L 123ff): “The results for the two resistance outcomes are qualitatively comparable in the sense that individual estimates may differ, but show similar absence of evidence to support either the benefit, harm or equivalence of treating with a higher number of antibiotics. Therefore, our …”. More detailed discussion about differences in estimates can be found in the SI, when the estimates of emergence of resistance are presented (e.g. SI section 2.1).

Line 123: "Furthermore, a lower number of antibiotics performed better than a higher number if the compared treatment arms had no antibiotics in common (pooled OR 4.73, 95% CI 2.14 - 10.42; I 2 = 37%, SI p 7).".How do you explain this? What does this mean?

We now added a more detailed explanation in the supplement (L 376ff.): “The result that if the treatment arms had no antibiotics in common a lower number of antibiotics performed better than a higher number of antibiotics could be due to different potencies of antibiotics or resistance prevalences. Further, there could be a bias to combine less potent antibiotics or antibiotics with higher resistance prevalence to ensure treatment efficacy, which couldlead to higher chances to detect resistances in the treatment arm with higher number of antibiotics, e.g. by selecting pre-existing resistance due to antibiotic treatment (see also section 6.1.9).” We furthermore already specifically mention this point in the main manuscript and refer then to the detailed explanation in the SI (L134 ff, “which could be due to different potencies or resistance prevalences of antibiotics as discussed in SI (SI section 6.1.10)”)

Overall, we want to point out that these results need to be interpreted with caution as overall the statistical power is limited to confidently estimate the difference in effect of a higher and lower number of antibiotics.

Line 125: ". In contrast, when restricting the analysis to studies with at least one common antibiotic in the treatment arms are pooled there was little evidence of a difference pooled OR 0.55, 95% CI 0.28 - 1.07".The difference was not statistically significant but there does seem to be an indication of a difference, please rephrase.

We rephrased the sentence to (L135 ff.): “In contrast, when restricting the analysis to studies with at least one common antibiotic in the treatment arms we found no evidence of a difference, only a weak indication that a higher number of antibiotics performs better (pooled OR 0.55, 95% CI 0.28 – 1.07; *I2* = 74%, figure 3B).”

Line 190: "Similarly, today, relevant cohort studies could be analysed collaboratively using various modern statistical methods to address confounding by indication and other biases (66, 67)".However, residual confounding by indication is likely. Please also mention the disadvantages of observational studies compared to RCTs.

We now highlight that causal inference with observational data comes with its own challenges and stress that randomized controlled trials are still considered the gold standard. L 204ff now reads: “However, even with appropriate causal inference methods, residual confounding cannot be excluded when using observational data (67). Therefore, will remain the gold standard to estimate causal relationships.”

Line 230: "Gram-negative bacteria have an outer membrane, which is absent in grampositive bacteria for instance, therefore intrinsic resistance against antibiotics can be observed in gram-negative bacteria (11)".Intrinsic resistance is not unique for Gram-negative bacteria but also exists for Grampositive bacteria.

We agree with the reviewer that intrinsic resistance is not unique to gram-negative bacteria and refined our writing. We additionally added that differences between gram-negative and gram-positive bacteria are not only to be expected due to differing intrinsic resistances but also due to potential differences in the mechanistic interactions of antibiotics, i.e., synergy or antagonism. The paragraph reads now (SI L289): “The gram status of a bacterium may potentially determine how effective an antibiotic, or an antibiotic combination is. Differences between gram-negative and gram-positive bacteria such as distinct bacterial surface organisation can lead to specific intrinsic resistances of gram-negative and grampositive bacteria against antibiotics (55). These structural differences can lead to varying effects of antibiotic combinations between gram-negative and gram-positive bacteria (56).”